# MANF regulates hypothalamic control of food intake and body weight

Su Yang[1], Huiming Yang[1,2], Renbao Chang[1], Peng Yin[1], Yang Yang[1,2], Weili Yang[3], Shanshan Huang[1,4], Marta A. Gaertig[1], Shihua Li[1,3] & Xiao-Jiang Li[1,3]

The hypothalamus has a vital role in controlling food intake and energy homeostasis; its activity is modulated by neuropeptides and endocrine factors. Mesencephalic astrocyte-derived neurotrophic factor (MANF) is a neurotrophic factor that is also localized in the endoplasmic reticulum (ER) in neurons. Here we show that MANF is highly enriched in distinct nuclei of the mouse hypothalamus, and that MANF expression in the hypothalamus is upregulated in response to fasting. Increasing or decreasing hypothalamic MANF protein levels causes hyperphagia or hypophagia, respectively. Moreover, MANF triggers hypothalamic insulin resistance by enhancing the ER localization and activity of PIP4k2b, a kinase known to regulate insulin signaling. Our findings indicate that MANF influences food intake and body weight by modulating hypothalamic insulin signaling.

[1] Department of Human Genetics, Emory University School of Medicine, 615 Michael Street, Room 355, Atlanta, GA 30322, USA. [2] Department of Neurology, Xiangya Hospital, Central South University, Changsha 410008, China. [3] GHM Institute of CNS Regeneration, Jinan University, Guangzhou 510631, China. [4] Department of Neurology, Tongji Hospital, Tongji Medical College, Huazhong University of Science and Technology, Wuhan 430032, China. Correspondence and requests for materials should be addressed to S.Y. (email: syang33@emory.edu) or to S.L. (email: sli@emory.edu) or to X.-J.L. (email: xli2@emory.edu)

Obesity is a global pandemic and a major risk factor for cardiovascular diseases, stroke, type 2 diabetes, high blood pressure and certain types of cancer[1], imposing profound economic and health care burdens on both individuals and society. Obesity is caused by unbalanced energy intake and expenditure, both of which are controlled by complex neuronal activities. The hypothalamus is a brain region known to regulate metabolic processes[2–4]. Several neuropeptides, such as orexin[5] and neuropeptide Y (NPY)[6], neurotrophic factors, such as brain-derived neurotrophic factor (BDNF)[7], and endocrine peptides, such as insulin and leptin[8], regulate food intake and body weight by altering the activities of several distinct hypothalamic neuronal populations. Disrupting the functions of these factors in the hypothalamus results in hyperphagia and obesity in rodent models[9–11], indicating neuronal activities mediated by signaling molecules play a pivotal role in maintaining energy homeostasis. Given that intracellular signaling in hypothalamic neurons is governed by multiple neurotrophic factors and peptides, additional molecules and mechanisms are likely involved in the hypothalamic regulation of food intake and metabolism.

Mesencephalic astrocyte-derived neurotrophic factor (MANF) is a recently identified neurotrophic factor that is structurally unrelated to the classical neurotrophic factor family[12]. Uniquely, MANF has two mechanisms of action: first, MANF is secreted into extracellular space, and addition of MANF protein protects specific types of neurons both in vitro and in vivo[13, 14], which suggests that like classical neurotrophic factors, MANF is able to modulate cellular activities by binding to an unknown receptor in the cell membrane; second, MANF is a soluble endoplasmic reticulum (ER) protein induced by the unfolded protein response (UPR), and is able to suppress apoptosis through its C-terminal domain[15, 16], indicating MANF possesses biological functions inside the cells as well. To date, MANF has been shown to be protective in several disease conditions, including Parkinson's disease, spinocerebellar ataxia 17, ischemic stroke and retinal degeneration[14, 17–21]. However, little is known about the endogenous functions of MANF in the brain. According to a recent study, the expression of MANF is highest during early postnatal days in different regions of rat brain, and gradually declines as the brain matures. Nonetheless, the high expression level of MANF in the hypothalamus persists into adulthood[22], raising the possibility that MANF plays an important role in the mature hypothalamus.

In the present study, we find that MANF is enriched in several hypothalamic nuclei that critically regulate energy intake, and the expression of MANF is increased in the hypothalamus of mice upon fasting. Increasing MANF expression in the hypothalamus lead to the development of hyperphagia and obesity, whereas reducing MANF expression in the hypothalamus leads to hypophagia and retarded body weight gain. Moreover, we identify that phosphatidylinositol 5-phosphate 4-kinase type-2 beta (PIP4k2b) is an interacting partner of MANF in ER and that MANF increases the localization of PIP4k2b in ER to mediate insulin resistance. These studies indicate MANF is involved in the hypothalamic control of food intake and energy homeostasis through mediating insulin signaling.

## Results

**Fasting increases the expression of MANF in the hypothalamus**. Using immunohistochemistry to examine the expression of MANF in the brain of wild type (WT) mice, we found that MANF is widely distributed in different brain regions. Nonetheless, the most intensive staining is seen in various hypothalamic regions, including the ARC and LH (Fig. 1a, Supplementary Fig. 1a), which critically regulate energy intake. A similar expression pattern of MANF in the hypothalamus was found

using another MANF antibody, but this expression pattern was lost in MANF knockdown mice using CRISPR/Cas9 (Supplementary Fig. 1b, details about the knockdown mice are described in later paragraph), indicating that MANF immunostaining is specific. This result was further confirmed by western blotting showing that, among different brain regions examined, hypothalamus has the highest level of MANF expression (Supplementary Fig. 1c). In addition, double immunostaining revealed that MANF was predominantly expressed in the neurons, but not in the astrocytes of the hypothalamus (Supplementary Fig. 1d). These results indicate that MANF may function in the hypothalamus to modulate energy homeostasis. To test this hypothesis, we fasted WT mice for 48 h, and checked MANF expression level by western blotting. Compared with ad libitum-fed littermates, fasted mice showed a significantly increased level of MANF in the hypothalamus (Fig. 1b, c). 4 h re-feeding of the fasted mice led to a rapid decrease of MANF in the hypothalamus (Supplementary Fig. 1e), suggesting that MANF level in the hypothalamus is closely linked to the feeding status. In contrast, the level of MANF was not changed in other brain regions, such as cortex, or non-neuronal tissues, such as liver (Fig. 1b, c). MANF increase in the hypothalamus is also supported by immunohistochemistry, as MANF staining intensity was increased in the fasted mice (Fig. 1d, e). By performing quantitative real-time PCR, we found a significantly increased level of MANF mRNA specifically in the hypothalamus of fasted mice (Fig. 1f). Altogether, these data suggest MANF is abundantly expressed in distinct hypothalamic nuclei that are important for energy homeostasis, and changes in energy balance lead to altered expression of MANF in the hypothalamus.

**Characterization of MANF transgenic mouse model**. If MANF is indeed involved in the hypothalamic control of energy homeostasis, changing its expression level in mice should lead to altered energy metabolism. We have generated a MANF transgenic mouse model, in which mouse *Manf* cDNA is linked with a sequence encoding HA tag at the 3′ end, under the control of mouse prion promoter[17]. Mouse prion promoter enables transgene expression in the central nervous system, preferentially in the neurons[23, 24]. Although quantitative real-time PCR results indicate a marked increase of MANF mRNA level in the brain of MANF transgenic mice compared with WT mice (Fig. 2a), western blotting with anti-MANF showed modest expression of transgenic MANF versus endogenous MANF in the brain of MANF transgenic mice (Fig. 2b). We also used anti-HA to define the expression of transgenic MANF. Transgenic MANF is expressed in the cortex and hypothalamus, but is not detected in the liver (Fig. 2c). Within the hypothalamus, transgenic MANF is predominantly expressed in the neurons, but not in the astrocytes (Supplementary Fig. 2a).

Interestingly, MANF transgenic mice fed with a regular chow diet become obese (Fig. 2d). Starting from 4-month of age, both male and female MANF transgenic mice displayed significantly increased body weights compared with their WT littermates (Fig. 2e). This result is unlikely due to disruption of genes that are essential for energy homeostasis caused by random transgene insertion, as another MANF transgenic line showed a similar phenotype (Supplementary Fig. 2b). The increased body weight correlates with increased adiposity (Fig. 2f), which was demonstrated by a significantly higher amount of gonad fat tissues in both male and female MANF transgenic mice (Fig. 2g). Nonetheless, the snout-anus length of MANF transgenic mice was comparable to WT mice (Fig. 2h).

The development of obesity could be due to increased energy intake, reduced energy expenditure, or both. Starting from 2-

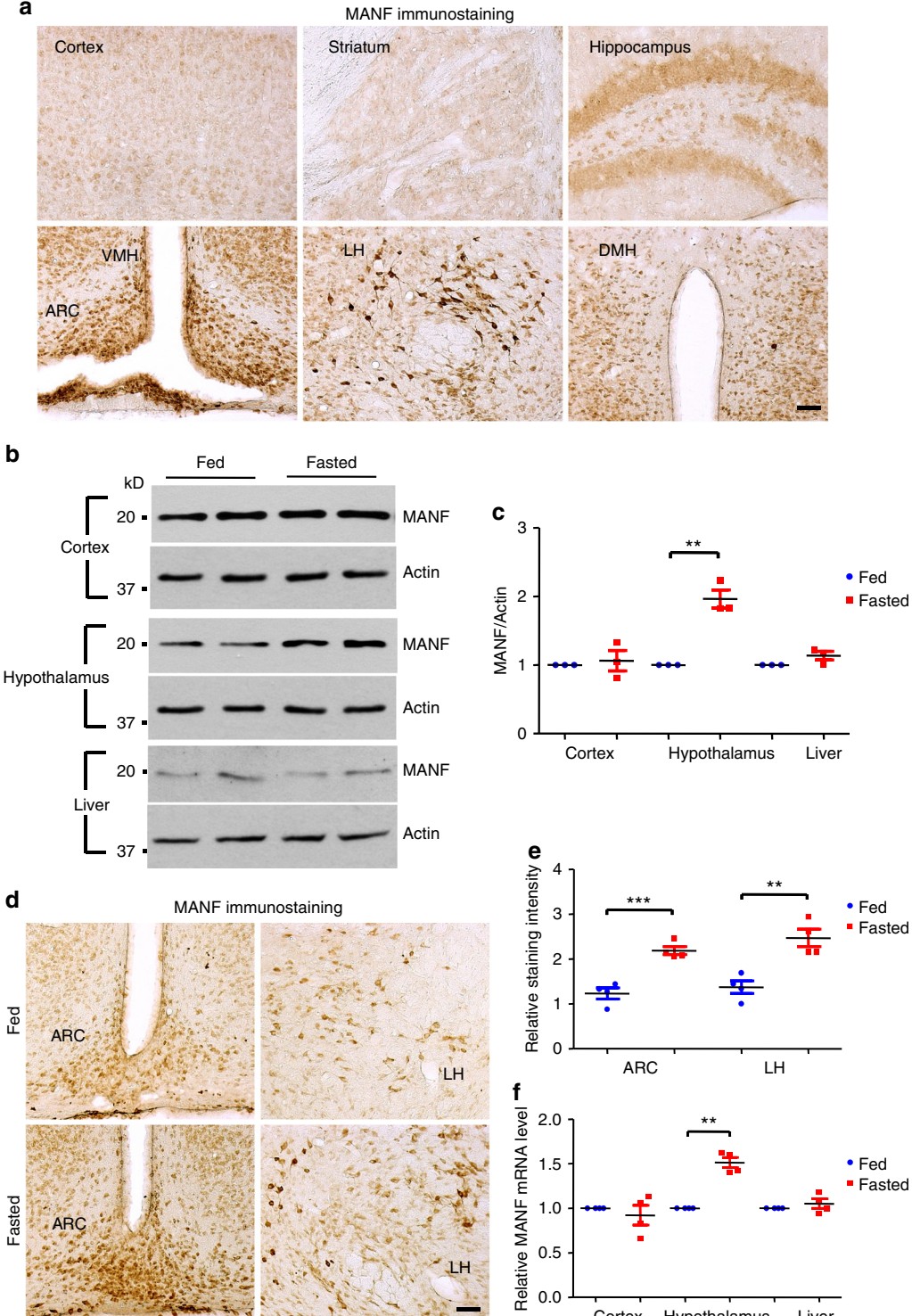

**Fig. 1** MANF is enriched in the hypothalamus and upregulated by energy deprivation. **a** Immunohistochemistry study revealed the expression of MANF in different regions of the mouse hypothalamus, including arcuate nucleus (ARC), ventromedial hypothalamus (VMH), dorsomedial hypothalamus (DMH) and lateral hypothalamus (LH) (*Scale bar*: 50 μm). **b** Representative western blotting images (from three independent experiments) showed that 48 h fasting increased MANF expression specifically in the mouse hypothalamus, but not in the cortex or liver. **c** Quantification of western blotting results in Fig. 1b (**$P < 0.01$, $n = 3$, 2-tailed student $t$ test, $t = 7.316$, $P = 0.0091$). **d** Immunohistochemistry showed that 48 h fasting increased MANF staining in the hypothalamus of wild type mice (*Scale bar*: 50 μm). **e** Quantification of MANF staining intensity in Fig. 1d (**$P < 0.01$, ***$P < 0.001$, $n = 4$; 2-tailed student $t$ test, ARC, $t = 6.213$, $P = 0.0008$; LH, $t = 4.569$, $P = 0.0038$). **f** qRT-PCR result indicated MANF mRNA level was significantly increased in the hypothalamus after 48 h fasting (**$P < 0.01$, $n = 4$, 2-tailed student $t$ test, Cortex, $t = 0.725$, $P = 0.5209$; Hypothalamus, $t = 8.891$, $P = 0.003$; Liver, $t = 0.9839$, $P = 0.3977$). Data are represented as mean ± SEM. Whenever you refer to a student t-test in a figure legend, please state whether test was 1-tailed or 2-tailed.

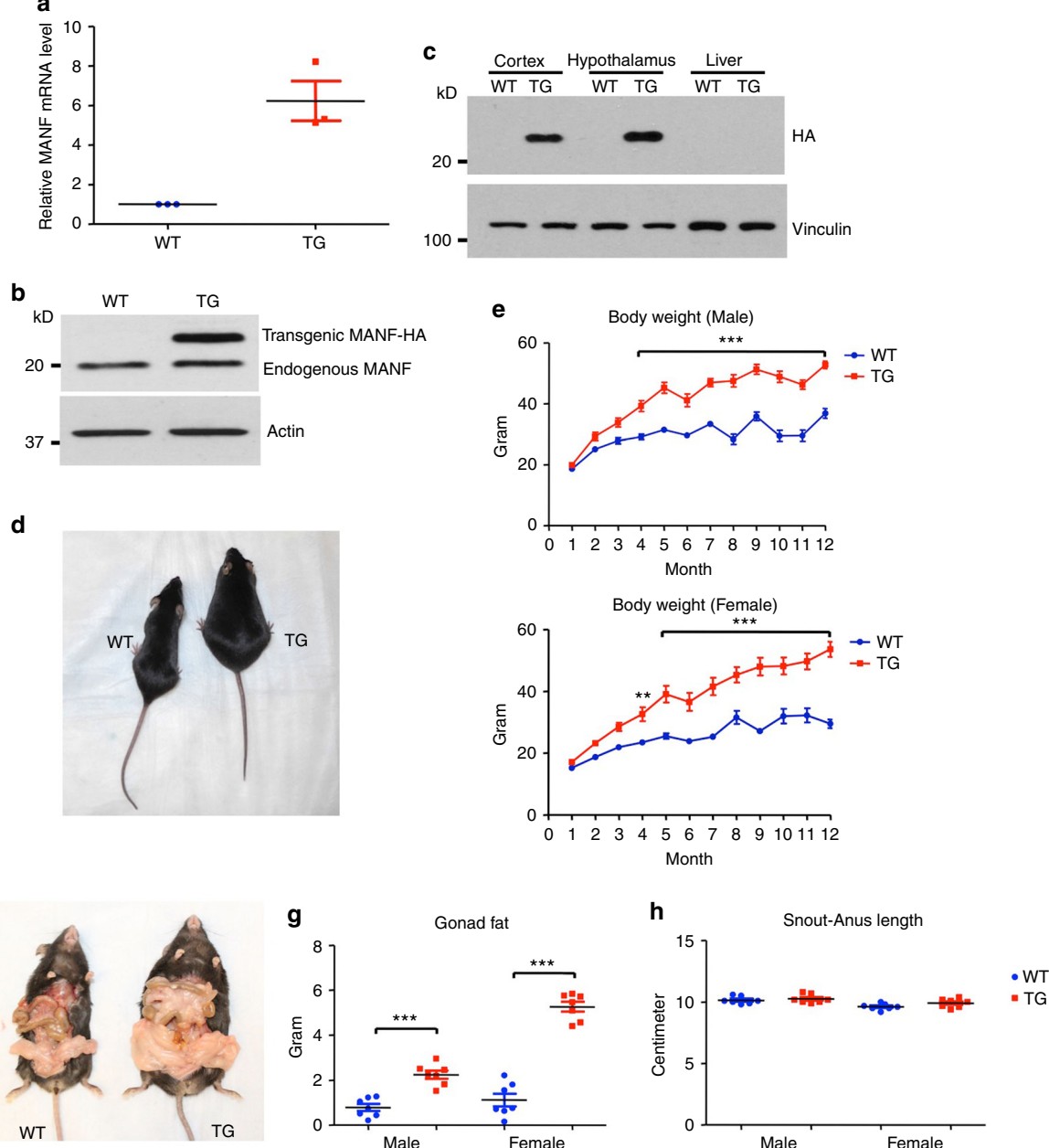

**Fig. 2** Characterization of MANF transgenic mouse model. **a** qRT-PCR result compared the levels of MANF mRNA in the brain of wild type (WT) and MANF transgenic (TG) mice ($n = 3$, 2-tailed student $t$ test, $t = 5.224$, $P = 0.0347$). **b** A single western blotting image showing endogenous and transgenic MANF expression in the brain of WT and TG mice. **c** A single western blotting image with an HA antibody confirmed that transgenic MANF is highly expressed in the cortex and hypothalamus of TG mice. **d** A single image of 4-month old WT and TG mouse. **e** Body weights of male and female mice were measured monthly from 1-month to 12-months of age (**$P < 0.01$, ***$P < 0.001$, $n = 8$–10, two-way ANOVA with Bonferroni post-tests, Male, F = 401.4, $P < 0.0001$; Female, F = 292.5, $P < 0.0001$). **f** A single image of 4-month-old WT and TG mice showing fat deposition. **g** Gonad fat of 4-month old male and female mice were weighed (***$P < 0.001$, $n = 7$; 2-tailed student $t$ test, Male, $t = 6.17$, $P < 0.0001$; Female, $t = 11.53$, $P < 0.0001$). **h** Snout-anus lengths of 4-month old male and female mice were measured $n = 7$, 2-tailed student $t$ test, Male, $t = 0.651$, $P = 0.5273$; Female, $t = 1.807$, $P = 0.0958$). Data are represented as mean ± SEM

month, MANF transgenic mice showed significantly increased food and water intake compared with WT littermates (Fig. 3a, Supplementary Fig. 3a). To determine if reduced energy expenditure also contributes to the obese phenotype in MANF transgenic mice, we used indirect calorimetry and found comparable levels of energy expenditure, oxygen consumption and respiratory exchange ratio between 3-month old WT and MANF transgenic mice (Fig. 3b–d). To exclude the potential influence of body weight variation on energy expenditure, we

performed the ANCOVA statistical test, which uses multiple linear regression analysis to assess the impact of body weight on energy expenditure[25–27]. The results indicate that the interaction between body weight and genotype was not significant, and energy expenditure between WT and MANF transgenic mice was not significantly different either (Supplementary Fig. 3b). Furthermore, we pair fed 6-week old MANF transgenic mice to restrict their energy intake to the same level of their WT littermates, until the mice reached 6-month old, at which point

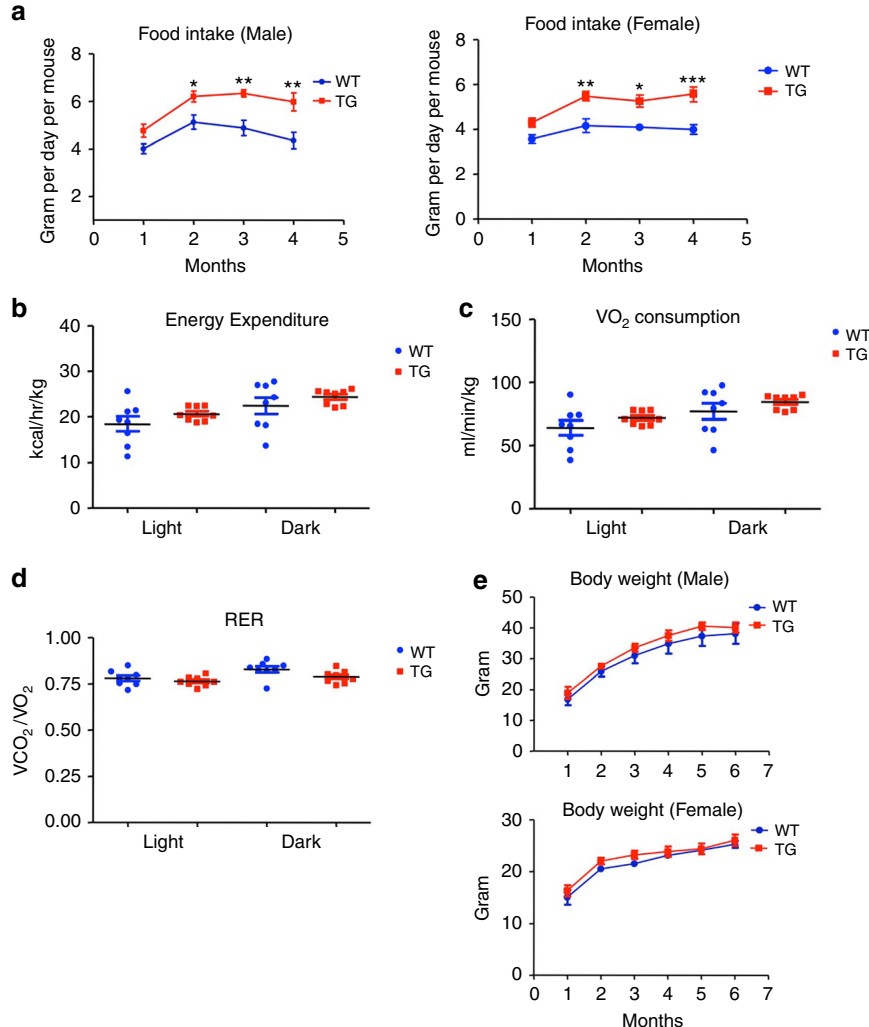

**Fig. 3** MANF transgenic mice exhibited increased food intake. **a** Daily food intake of wild type (WT) and MANF transgenic (TG) mice were measured monthly from 1-month to 4-months of age (*$P < 0.05$, **$P < 0.01$, $n = 7–8$, two-way ANOVA with Bonferroni post-tests, Male, F = 37.8, $P < 0.0001$; Female, F = 46.95, $P < 0.0001$). **b** Energy expenditure of 3-month old WT and TG mice was measured ($n = 8$, 2-tailed student $t$ test, Light, $t = 1.226$, $P = 0.2261$; Dark, $t = 1.02$, $P = 0.3248$). **c** Rate of oxygen consumption of 3-month old WT and TG mice was measured ($n = 8$, 2-tailed student $t$ test, Light, $t = 1.256$, $P = 0.2295$; Dark, $t = 1.104$, $P = 0.288$). **d** Respiratory exchange ratio of 3-month old male and female WT and TG mice was measured ($n = 8$, 2-tailed student $t$ test, Light, $t = 0.9832$, $P = 0.3422$; Dark, $t = 2.001$, $P = 0.0651$). **e** Starting from the age of 6 weeks, TG mice were pair-fed to WT littermates, and the body weights of WT and TG mice were measured monthly from 1-month to 6-months of age ($n = 5$, two-way ANOVA with Bonferroni post-tests, Male, F = 3.558, $P = 0.9994$; Female, F = 4.225, $P = 0.9604$). Data are represented as mean ± SEM

MANF transgenic mice are supposed to become highly obese. The body weights of both male and female pair-fed MANF transgenic mice were similar to those of WT littermates during the test period (Fig. 3e). We also monitored the locomotor activity of 2-month old mice over a 24-h period, and MANF transgenic mice displayed a similar level of activity as WT littermates (Supplementary Fig. 3c). These results indicate that increased energy intake is the sole cause of obesity in MANF transgenic mice.

**Hypothalamic MANF overexpression leads to hyperphagia.** The fact that MANF is highly enriched in the hypothalamus and transgenic MANF causes obesity in MANF transgenic mice leads us to propose MANF functions in the hypothalamus to regulate energy homeostasis. However, transgenic MANF is widely expressed in the brain, making it difficult to study the role of MANF specifically in the hypothalamus. To circumvent this difficulty, we generated adeno-associated virus expressing MANF

with a C-terminal HA tag (AAV-MANF) under the control of ubiquitin C (UBC) promoter (Fig. 4a). We used stereotaxic instruments to inject this virus (1 µl in each side) into the hypothalamus of 2-month old WT mice and verified that most areas of the hypothalamus were infected. As a control, we injected another adeno-associated virus expressing GFP (AAV-GFP) into the same location of the hypothalamus of 2-month old WT littermates. By western blotting with an HA antibody, we were able to detect the robust expression of viral MANF specifically in the hypothalamus (Fig. 4b). By immunofluorescent staining, we confirmed that viral MANF expression was widely distributed in different nuclei of the hypothalamus, including the ARC, LH, and dorsomedial hypothalamus (DMH) (Fig. 4c, *left* panel). Moreover, virally expressed MANF is predominantly localized in the cytoplasm of infected cells (Fig. 4c, *right* panel), which is the same localization as endogenous MANF. Interestingly, both male and female WT mice injected with AAV-MANF showed a rapid and progressive onset of body weight increase (Fig. 4d). As early as 2 weeks post injection, their body weights became significantly

higher than WT littermates injected with AAV-GFP (Fig. 4e), although their snout-anus lengths are indistinguishable (Supplementary Fig. 4a). In agreement with the body weight increase, WT mice injected with AAV-MANF exhibited significant hyperphagia (Fig. 4f, Supplementary Fig. 4b). We also injected a lower dose (0.2 μl in each side) of AAV-MANF to specifically target the ARC or LH (Supplementary Fig. 4c). Injection into the ARC, but not the LH, led to a modest increase in food intake and

body weight, but this increase was significantly less than that with whole hypothalamus infection (Supplementary Fig. 4d, e). This result suggests that the ARC and other hypothalamic regions are required for the robust effect of MANF on food intake and body weight. MANF possesses both intracellular and extracellular activities. We injected recombinant mouse MANF protein either into the third ventricle (10 μg) or directly into the hypothalamus (8 μg in each side) of WT mice, which would act on hypothalamic

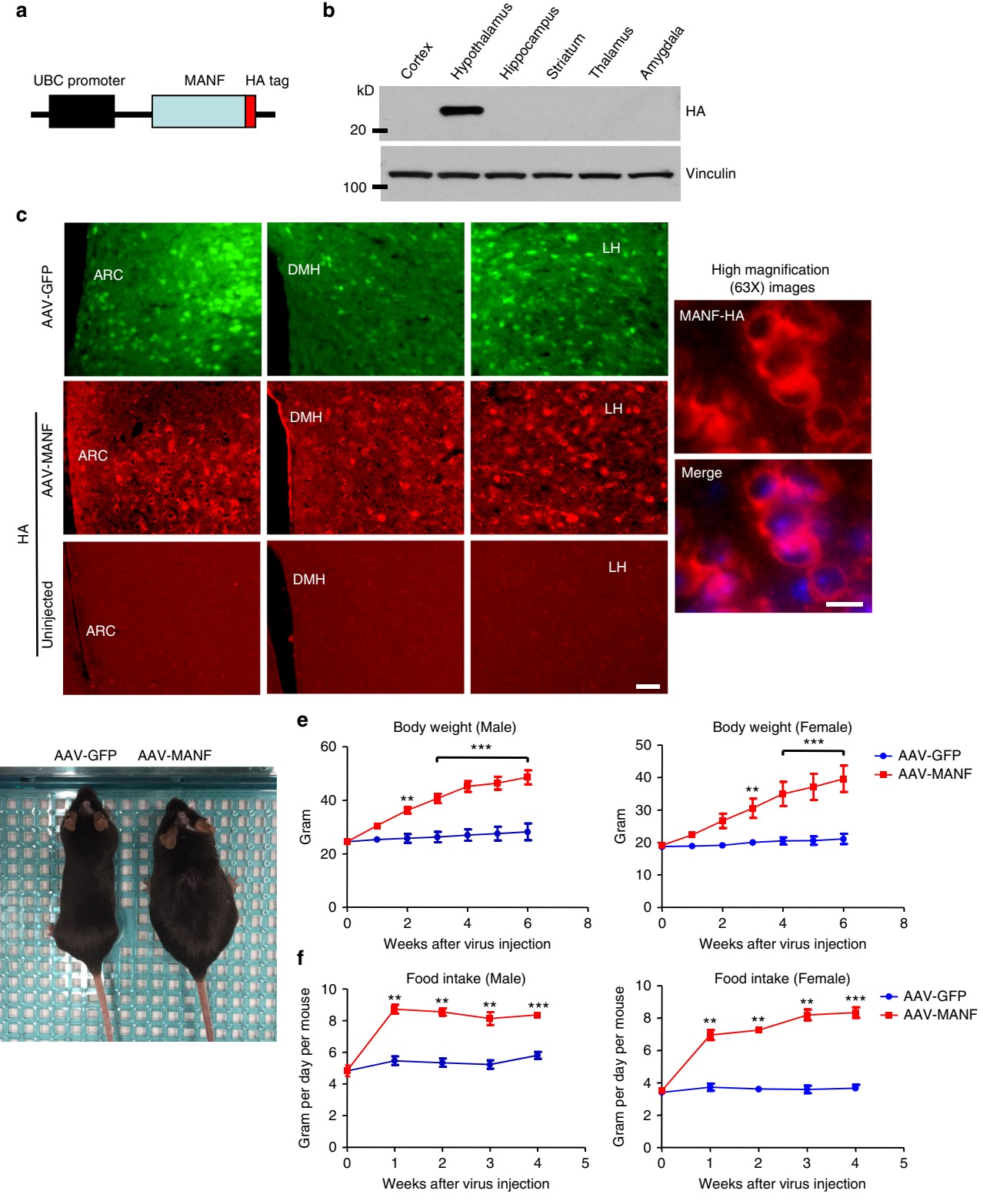

neurons extracellularly. However, we did not find any changes in food intake, compared with control WT mice injected with PBS (Supplementary Fig. 4f, g). Therefore, MANF is more likely to function intracellularly in the hypothalamus to regulated food intake.

**Hypothalamic reduction of MANF leads to hypophagia.** Next, we knocked down MANF in the hypothalamus using an adeno-associated virus expressing MANF shRNA (AAV-MANF-shRNA) (Fig. 5a). AAV-MANF-shRNA led to a significant reduction of MANF in the hypothalamus, detected by western blotting (Fig. 5b, c). One week after viral injection, the daily food intake of WT mice injected with AAV-MANF-shRNA was significantly reduced, compared with littermates injected with AAV scramble shRNA (Fig. 5d). Those mice also displayed retarded body weight gain (Fig. 5e). To knock down MANF expression in a cell type-specific manner, we made use of CRISPR system, with Cas9 under the control of the neuron-specific Mecp2 promoter[28], or astrocyte-specific Gfap promoter (Fig. 5f). The targeting efficacy of designed sgRNA to *Manf* was verified by sequencing (Fig. 5g) and T7 endonuclease 1 assay (Supplementary Fig. 5a). Co-injection of viruses encoding MANF sgRNA (AAV-MANF-sgRNA) and neuron-specific Cas9 (AAV-Mecp2-Cas9) knocked down > 80% of MANF in the hypothalamus (Fig. 5h, i, Supplementary Fig. 1a), again demonstrating that MANF is predominantly expressed in the neurons. Importantly, WT mice with neuron-specific deletion of MANF had significantly reduced food intake and retarded body weight gain, compared with littermates co-injected with AAV-control-sgRNA and AAV-Mecp2-Cas9 (Fig. 5j, k). In contrast, co-injection of AAV-MANF-sgRNA and astrocyte-specific Cas9 (AAV-Gfap-Cas9) did not lead to a significant reduction of MANF in the hypothalamus, nor did it change food intake or body weight (Supplementary Fig. 5b–d). Therefore, decreasing MANF expression specifically in hypothalamic neurons, but not in astrocytes, reduces food intake and body weight gain.

**Hypothalamic MANF overexpression leads to insulin resistance.** Hypothalamic insulin signaling plays a pivotal role in regulating food intake and energy homeostasis[2, 3, 29–31]. Disrupting insulin signaling in the central nervous system[10] or in the hypothalamus[32] led to hyperphagia and increased fat mass. To investigate if hyperphagia activity in MANF transgenic mice is caused by impaired insulin signaling in the hypothalamus, we fasted 2-month old WT and MANF transgenic mice for 24 h; gave them an intraperitoneal injection of insulin (2 mg kg$^{-1}$ body weight); sacrificed them 45 min post insulin injection. We used a phospho-AKT (Ser473) antibody to detect insulin signaling intensity. Of all the tissues examined, which include the cortex, hypothalamus and liver, WT mice injected with insulin showed a greatly enhanced AKT phosphorylation level, suggesting a normal insulin response. In contrast, the cortex and hypothalamus, but not the liver, from MANF transgenic mice injected with insulin

had a significantly reduced AKT phosphorylation level (Fig. 6a, b), which indicates insulin resistance. We further confirmed this result using 2-month old WT mice injected with AAV-MANF in the hypothalamus. 2 weeks after viral injection, the mice were fasted and injected with insulin as described above. In the hypothalamus of WT mice injected with AAV-MANF, where viral MANF was abundantly expressed, the AKT phosphorylation upon insulin treatment was greatly diminished compared with WT mice injected with AAV-GFP. In the white adipose tissue, viral MANF was not detected, and the AKT showed comparable levels of phosphorylation upon insulin treatment (Fig. 6c, d). To confirm that insulin resistance in the hypothalamus caused increased food intake, we injected insulin (1 µU) into the third ventricle of WT and MANF transgenic mice. WT mice had a 60% reduction of daily food intake after insulin injection, whereas MANF transgenic mice had a 30% reduction (Supplementary Fig. 6a), suggesting that MANF transgenic mice were less sensitive to the food intake regulation mediated by insulin. Hypothalamic insulin signaling also regulates systemic glucose homeostasis[32–34], we found significantly elevated blood glucose, and impairment of glucose tolerance and insulin tolerance in 6-week old MANF transgenic mice (Fig. 6e–g). These mice also exhibit age-related liver steatosis (Supplementary Fig. 6b). Insulin stimulates the expression of proopiomelanocortin (POMC), a well known anorexigenic neuropeptide[35, 36]. In the hypothalamus of mice injected with AAV-MANF, we found a significantly reduced level of POMC (Supplementary Fig. 6c). We also compared insulin receptor (IRβ) expression in the hypothalamus of WT and MANF transgenic mice, and did not find any significant differences (Supplementary Fig. 6d). Leptin is another major signaling molecule for food intake regulation. In 6-week old WT and MANF transgenic mice, we found comparable levels of plasma leptin (Supplementary Fig. 6e). In addition, intraperitoneal injection of leptin (2 mg kg$^{-1}$ body weight) led to a similar level of STAT3 phosphorylation in the hypothalamus of WT and MANF transgenic mice (Supplementary Fig. 6f), indicating the leptin signaling was not affected by MANF overexpression.

There is growing evidence supporting that inflammation triggered by ER stress could lead to insulin resistance[37–39]. To rule out the possibility that transgenic MANF causes insulin resistance via enhanced inflammation, we checked the phosphorylation levels of several inflammatory molecules: IκB kinase α/β (IKK α/β) and c-Jun N-terminal kinase (JNK) in the hypothalamus of 2-month old WT and MANF transgenic mice. Both pIKK α/β and pJNK levels were comparable between WT and MANF transgenic mice (Supplementary Fig. 7a). Thus, transgenic MANF triggers insulin resistance via certain mechanisms other than inflammation. We also validated the inhibitory effects of MANF on insulin signaling in vitro. We transfected PC12 cells with pRK5 vector expressing MANF with a C-terminal HA tag (MANF-HA). Two days after transfection, PC12 cells were serum starved in DMEM medium for 4 h and treated with insulin (1 µg/ml) for different lengths of time. We found that insulin-induced

**Fig. 4** MANF overexpression in the hypothalamus led to increased body weight and food intake. **a** Schematic representation of AAV-MANF construct. **b** A single western blotting analysis of viral MANF expression in different brain regions of 2-month old wild type (WT) mice injected with AAV-MANF into the hypothalamus. **c** Immunofluorescence images of 2-month-old WT mice injected with AAV-GFP or AAV-MANF into the hypothalamus. Green fluorescence was used to indicate the expression of GFP in different regions of the hypothalamus, including arcuate nucleus (ARC), dorsomedial hypothalamus (DMH) and lateral hypothalamus (LH). Red fluorescence was used to indicate the expression of MANF, which was stained by an HA antibody. Higher magnification (63×) images revealed cytoplasmic localization of MANF. Nuclear staining is shown in the merged image. Staining of hypothalamus from uninjected mice was used as negative controls (*Scale bar*: *left* 50 µm, *right* 10 µm). **d** A single image of WT mice 2 weeks after AAV-GFP or AAV-MANF injection. **e** Body weights of male and female 2-month-old WT mice were measured weekly before and after AAV-GFP or AAV-MANF injection (*P < 0.05, **P < 0.01, n = 5, two-way ANOVA with Bonferroni post-tests, Male, F = 153.3, P < 0.0001; Female, F = 74.98, P < 0.0001). **f** Food intake of male and female 2-month-old WT mice were measured weekly before and after AAV-GFP or AAV-MANF injection (**P < 0.01, ***P < 0.001, n = 5, two-way ANOVA with Bonferroni post-tests, Male, F = 204.8, P < 0.0001; Female, F = 524.2, P < 0.0001). Data are represented as mean ± SEM

AKT phosphorylation was greatly repressed in cells transfected with MANF, compared with untransfected controls (Supplementary Fig. 7b). Moreover, by double staining with HA and pAKT antibodies, we confirmed that the individual cells overexpressing

MANF had low levels of phosphorylated AKT (Supplementary Fig. 7c). These results demonstrate that the overexpression of MANF in cultured cells leads to a reduced insulin sensitivity, which is consistent with our findings in vivo.

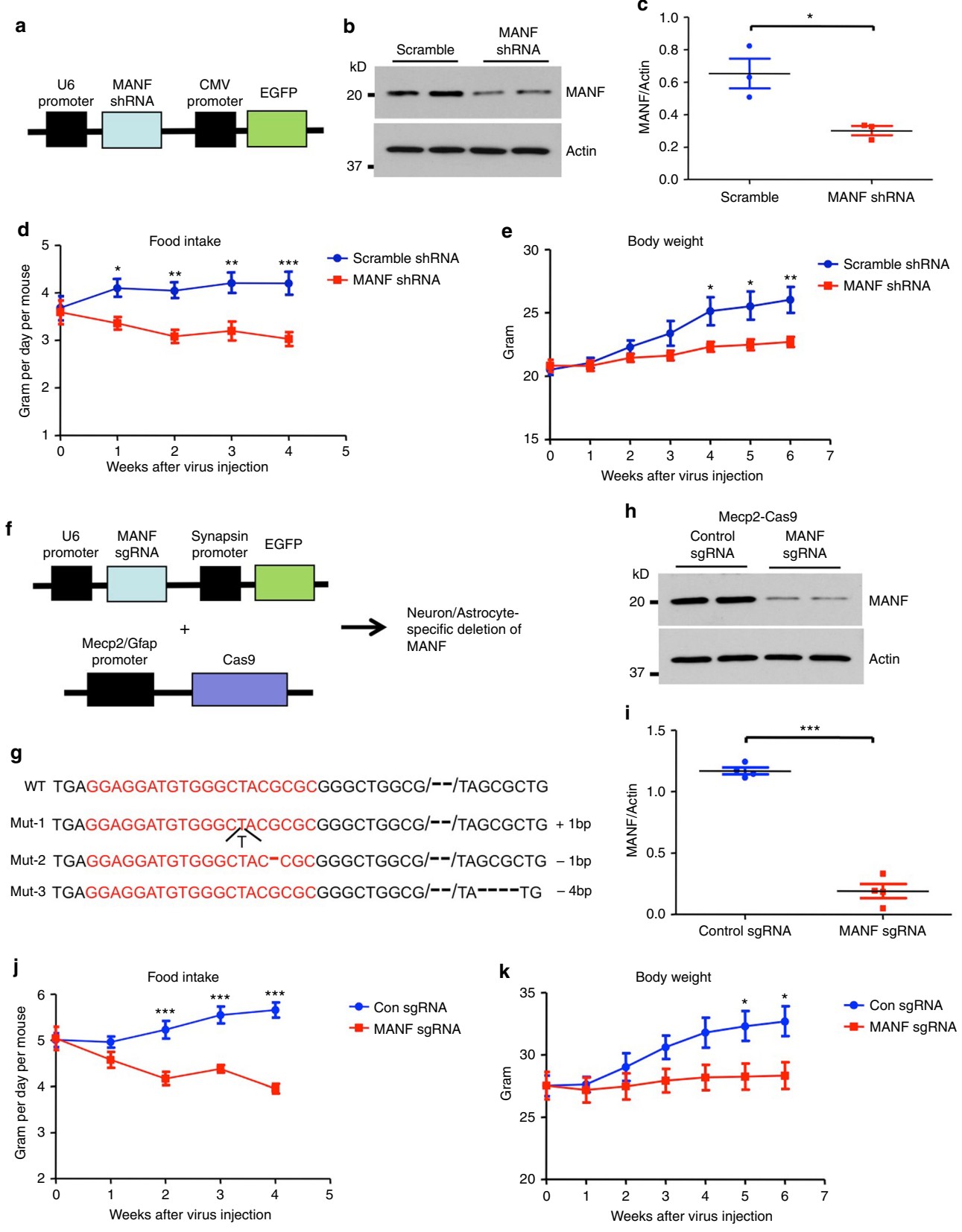

**MANF interacts with PIP4k2b**. Next we aimed to study the mechanisms underlying MANF-mediated insulin resistance by exploring its interacting partners via affinity-purification mass spectrometry. Previously we generated a bacterial expression construct that expresses mouse MANF with a C-terminal His tag (MANF-His)[17], which allows us to enrich recombinant MANF-His protein by Ni-NTA affinity chromatography. We mixed purified MANF-His protein with mouse brain lysate. After overnight incubation, Ni-NTA beads were added into the mixture to pull down MANF-His, together with MANF interacting proteins. As two negative controls, we performed the same procedure using brain lysate only or recombinant MANF-His protein only (Fig. 7a). The pull-down lysates from these three reactions were loaded to a Tris-Glycine gel for electrophoresis, and protein bands were revealed by silver staining. We found that one distinct band was selectively present in the pull-down lysate of brain lysate and MANF-His protein mixture (Fig. 7b). This band was excised and analyzed by mass spectrometry, and one protein identified from that band is phosphatidylinositol 5-phosphate 4-kinase type-2 beta (PIP4k2b). PIP4k2b catalyzes the phosphorylation of phosphatidylinositol-5-phosphate (PI-5-P) to form phosphatidylinositol-4,5-bisphosphate (PI-4,5-$P_2$)[40]. PI-4,5-$P_2$ is further phosphorylated by phosphoinositide 3-kinase (PI3K) to generate phosphatidylinositol-3,4,5-trisphosphate (PI-3,4,5-$P_3$). The expression of PIP4k2b reduced the level of PI-3,4,5-$P_3$, which led to decreased activation of AKT upon insulin treatment[41]. In addition, PIP4k2b knockout mice are hypersensitive to insulin and have reduced body weight compared with WT littermates[42]. These previous findings provided us a strong rational to investigate whether increased MANF augments PIP4k2b function to yield insulin resistance.

We carried out a series of experiments to confirm the interaction between MANF and PIP4k2b. First, we generated and purified GST tagged recombinant PIP4k2b protein (GST-PIP4k2b), which was incubated with MANF-His protein, and pulled down by glutathione beads. We were able to detect abundant MANF-His protein in the pull-down lysate (Fig. 7c). Next, we transfected PC12 cells with MANF-HA and Myc tagged PIP4k2b (Myc-PIP4k2b). By pulling down MANF-HA with an HA antibody, we found that Myc-PIP4k2b was co-immunoprecipitated (Fig. 7d). We also performed this co-immunoprecipitation assay using brain lysate of MANF transgenic mice, and endogenous PIP4k2b was detected in the pull-down lysate (Fig. 7e). To identify the protein domain within PIP4k2b that interacts with MANF, we generated three truncated GST tagged PIP4k2b fragments (Fig. 7f, *upper* panel). By in vitro binding assay, we found that the N-terminus of PIP4k2b (amino acid 1-126) was critical for its interaction with MANF (Fig. 7f, *lower* panel).

The synthesis of PI-4,5-$P_2$ occurs in the ER[43], and one isoform of PIP4k2 predominantly localizes in the ER to function[44]. Importantly, MANF is also enriched in the ER[15, 45], suggesting that MANF potentially interacts with PIP4k2b in the ER to regulate its functions. We performed subcellular fractionation using brain lysates from 2-month old WT and MANF transgenic mice. As expected, both PIP4k2b and transgenic MANF were highly enriched in the ER fraction. Interestingly, the amount of PIP4k2b in the ER fraction was greatly increased in the brain lysate of MANF transgenic mice (Fig. 7g). A closer examination revealed that the overall level of PIP4k2b did not change in the brain of MANF transgenic mice compared with WT littermates, but its level in the ER fraction was significantly elevated. Moreover, in the liver of MANF transgenic mice, where MANF is not overexpressed, the amount of PIP4k2b in the ER fraction remained comparable to WT controls (Fig. 7h, i). We further checked PIP4k2b expression in fasted WT mice, MANF transgenic mice with intraperitoneal injection of insulin, and WT mice with MANF knockdown, but did not find any changes in PIP4k2b protein levels (Supplementary Fig. 8).

A similar trend was observed in vitro, as PC12 cells transfected with MANF-HA showed a further enrichment of PIP4k2b in the ER fraction compared with untransfected controls (Supplementary Fig. 9a, b). As the N-terminus of PIP4k2b binds to MANF, we expressed a mutant version of PIP4k2b with an N-terminal truncation (amino acid 1-106) in PC12 cells. Deleting N-terminal PIP4k2b led to a dramatic reduction of PIP4k2b in the ER fraction (Supplementary Fig. 9c). Consequently, truncated PIP4k2b was not as efficient as full length PIP4k2b in inhibiting insulin-stimulated AKT phosphorylation (Supplementary Fig. 9d, e). Taken together, these data indicate MANF plays an important role in regulating the localization of PIP4k2b in the ER, where PIP4k2b functions to inhibit insulin signaling.

**Reducing PIP4k2b ameliorates insulin resistance**. If the enhanced ER localization and activity of PIP4k2b underlies MANF-mediated insulin resistance, reducing PIP4k2b should alleviate the impaired insulin response. To confirm this hypothesis, we used pooled PIP4k2b siRNA lentivirus to knock down PIP4k2b in PC12 cells transfected with MANF-HA. 2 days after transfection and viral infection, the cells were treated with insulin and the lysates were collected for western blotting analysis. Compared with control cells infected with scramble siRNA lentivirus, cells infected with PIP4k2b siRNA lentivirus had a reduced level of PIP4k2b, and the level of AKT phosphorylation was significantly increased (Fig. 8a, b). This result suggests that reducing PIP4k2b is able to improve insulin sensitivity in the presence of MANF overexpression in vitro.

**Fig. 5** Decreasing MANF in the hypothalamus reduces food intake. **a** Schematic representation of AAV-MANF-shRNA construct. **b** Representative western blotting images (from three independent experiments) of MANF expression in the hypothalamus of wild type (WT) mice injected with AAV-scramble-shRNA (Scramble) or AAV-MANF-shRNA (MANF shRNA). **c** Quantitative result of Fig. 5b (*$P < 0.05$, $n = 3$, 2-tailed student $t$ test, $t = 3.66$, $P = 0.0216$). **d** Daily food intake of 3-month old female WT mice was measured weekly before and after AAV-scramble-shRNA or AAV-MANF-shRNA injection (**$P < 0.01$, $n = 6$, two-way ANOVA with Bonferroni post-tests, F = 41.24, $P < 0.0001$). **e** Body weights of 3-month old female WT mice were measured weekly before and after AAV-scramble-shRNA or AAV-MANF-shRNA injection (*$P < 0.05$, **$P < 0.01$, $n = 6$, two-way ANOVA with Bonferroni post-tests, F = 21.4, $P < 0.0001$). **f** Schematic representation of AAV-MANF-sgRNA, AAV-Mecp2-Cas9 and AAV-Gfap-Cas9 constructs. **g** Sequencing results of *Manf* genomic locus targeted by sgRNA. Three representative mutant (mut) sequences after Cas9 cutting were shown, sgRNA sequence was highlighted in *red*. **h** Representative western blotting images (from three independent experiments) of MANF expression in the hypothalamus of wild type (WT) mice injected with AAV-control-sgRNA/AAV-Mecp2-Cas9 (Control sgRNA) or AAV-MANF-sgRNA/AAV-Mecp2-Cas9 (MANF sgRNA). **i** Quantitative result of Fig. 5h (***$P < 0.001$, $n = 4$, 2-tailed student $t$ test, $t = 15.30$, $P < 0.0001$). **j** Daily food intake of 3-month old male WT mice was measured weekly before and after AAV-control-sgRNA/AAV-Mecp2-Cas9 or AAV-MANF-sgRNA/AAV-Mecp2-Cas9 injection (***$P < 0.001$, $n = 6$, two-way ANOVA with Bonferroni post-tests, F = 70.06, $P < 0.0001$). **k** Body weights of 3-month old male WT mice were measured weekly before and after AAV-control-sgRNA/AAV-Mecp2-Cas9 or AAV-MANF-sgRNA/AAV-Mecp2-Cas9 injection (*$P < 0.05$, $n = 6$, two-way ANOVA with Bonferroni post-tests, F = 18.56, $P < 0.0001$). Data are represented as mean ± SEM

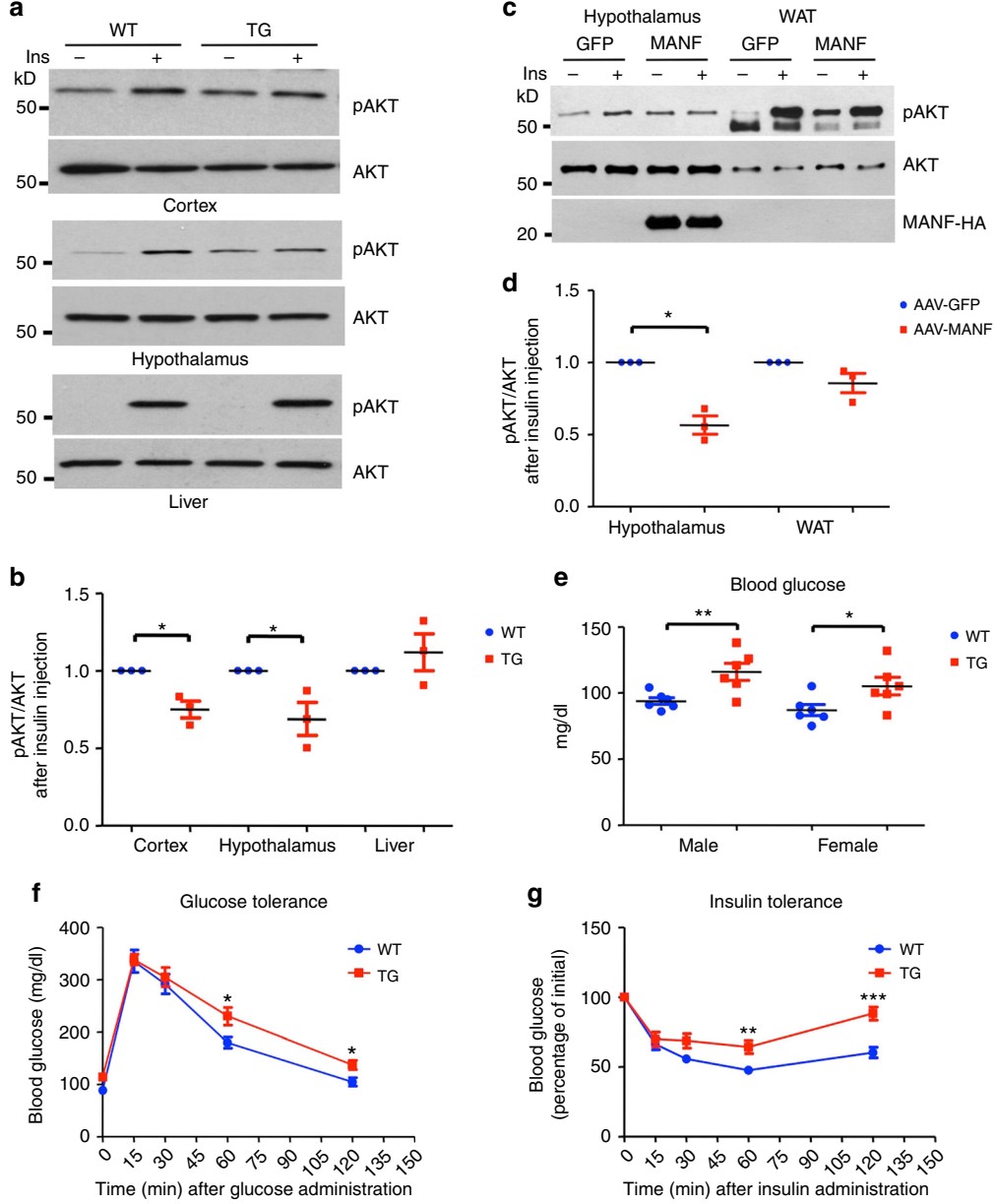

**Fig. 6** MANF overexpression leads to insulin resistance. **a** Representative western blotting images (from three independent experiments) of pAKT levels in wild type (WT) or MANF transgenic (TG) mice injected with either insulin (Ins+) or PBS (Ins−). **b** Quantification of western blotting result in Fig. 6a. Relative pAKT levels of WT and TG mice injected with insulin were calculated using the ratio of pAKT band intensity to AKT band intensity (*$P < 0.05$, **$P < 0.01$, $n = 3$; 2-tailed student $t$ test, Cortex, $t = 4.686$, $P = 0.0426$; Hypothalamus, $t = 2.941$, $P = 0.0494$; Liver, $t = 0.9982$, $P = 0.4233$). **c** Representative western blotting images (from three independent experiments) of pAKT levels in WT mice transduced with either AAV-GFP (GFP) or AAV-MANF (MANF) in the hypothalamus, and then injected with either insulin (Ins+) or PBS (Ins−). **d** Quantification of western blotting result in Fig. 6c (*$P < 0.05$, $n = 3$; 2-tailed student t test, Hypothalamus, $t = 6.905$, $P = 0.0203$; WAT, $t = 2.13$, $P = 0.1669$). **e** Blood glucose levels of 6-week old WT and TG mice were measured (*$P < 0.05$, **$P < 0.01$, $n = 6$; 2-tailed student $t$ test, Male, $t = 3.196$, $P = 0.0096$; Female, $t = 2.319$, $P = 0.0428$). **f** Glucose tolerance of 6-week old WT and TG mice was measured (*$P < 0.05$, $n = 8$; 2-tailed student $t$ test, 60-min, $t = 2.472$, $P = 0.0269$; 120-min, $t = 2.721$, $P = 0.0166$). **g** Insulin tolerance of 6-week old WT and TG mice was measured (*$P < 0.05$, $n = 6$; 2-tailed student $t$ test, 60-min, $t = 3.262$, $P = 0.0085$; 120-min, $t = 4.677$, $P = 0.0009$). Data are represented as mean ± SEM

Next, we tested this hypothesis in vivo by utilizing adeno-associated viruses encoding PIP4k2b shRNA (AAV-PIP4k2b-shRNA). Injection of either AAV scramble shRNA or AAV-PIP4k2b-shRNA led to widespread expression of viral constructs in the hypothalamus, evidenced by the GFP reporter (Fig. 8c) Interestingly, 3-month old MANF transgenic mice injected with AAV-PIP4k2b-shRNA into the hypothalamus had a significantly reduced food intake, compared with those injected with AAV scramble shRNA (Fig. 8d), which was accompanied by a

significant reduction of body weight (Fig. 8e). We also treated mice with insulin using the same method described above, and collected the cortex and hypothalamus tissues for western blotting. A decreased level of PIP4k2b was only found in the hypothalamus, as AAV-PIP4k2b-shRNA was delivered specifically to the hypothalamus. Consistently, we saw a significant increase of pAKT level in the hypothalamus of mice with AAV-PIP4k2b-shRNA (Fig. 8f, g). The expression of MANF protein was not affected by AAV-PIP4k2b-shRNA (Supplementary

Fig. 10). Altogether, these data indicate that reducing PIP4k2b is able to ameliorate the impaired hypothalamic insulin sensitivity and hyperphagia phenotype in MANF transgenic mice.

## Discussion

In summary, we identified MANF as a non-classical neurotrophic factor involved in the hypothalamic control of food intake and energy balance. MANF achieves its regulatory functions by mediating insulin signaling through its interaction with PIP4k2b in the ER. MANF can recruit PIP4k2b into the ER, where PIP4k2b becomes active and reduces the activation of AKT downstream of insulin pathway (Fig. 8h). Therefore, increasing MANF level in the hypothalamus leads to an impaired insulin response, which results in hyperphagia and obesity.

We found strong evidence supporting the involvement of MANF in controlling food intake through the hypothalamus.

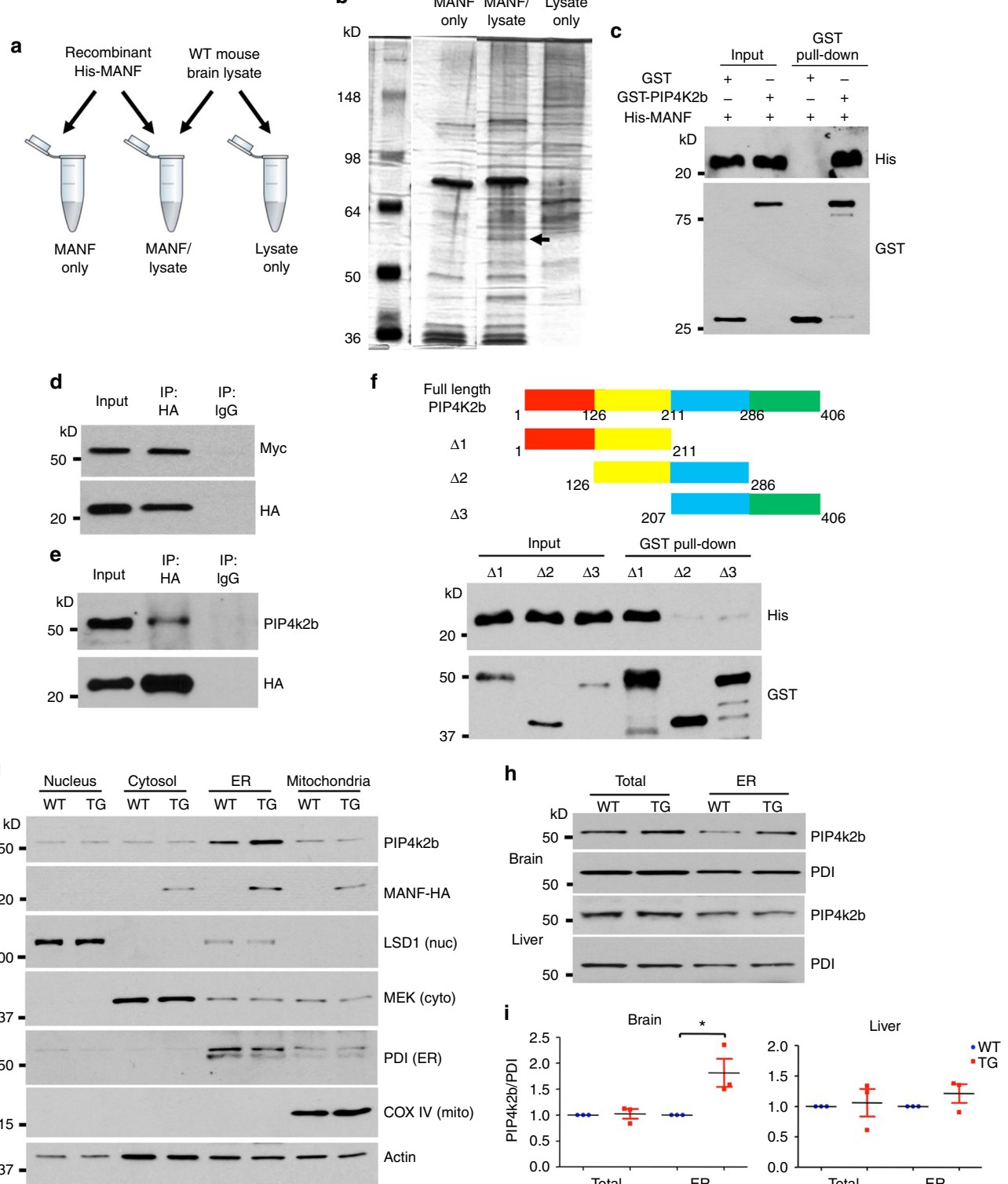

First, MANF is highly enriched in the hypothalamus, especially in the nuclei critical for the regulation of food intake. Second, the hypothalamic level of MANF is dependent on the energy status, which is demonstrated by the increase of MANF mRNA and protein, specifically in the hypothalamus of mice with energy restriction. Third, MANF transgenic mice have increased food intake and body weight gain, which is corroborated by a similar finding in WT mice with hypothalamic injection of MANF virus. Fourth, we found an opposite effect in food intake activity once MANF expression is specifically reduced in the hypothalamus. Recently, a MANF knockout (KO) mouse model was generated, and the mice developed a severe postnatal phenotype caused by reduction of β cells[46]. Germline deletion of MANF and extremely shortened life span in this KO mouse model make it difficult to assess the hypothalamus-specific loss of MANF function in the adult animals. Nonetheless, MANF KO mice display retarded growth and reduced fat deposition, which supports the opposite phenotypes in MANF transgenic mice.

Insulin plays an essential role in the maintenance of metabolic homeostasis by promoting the storage of energy in its primarily responsive organs, including the liver, fat and muscle. However, several lines of evidence also indicate insulin critically regulates energy intake by mediating neuronal activities in the central nervous system (CNS), particularly in the hypothalamus. Direct infusion of exogenous insulin to the hypothalamus reduces food intake[30, 31], and inhibition of insulin signaling in the hypothalamus leads to hyperphagia[32]. Interestingly, knocking out insulin receptor in the CNS led to a moderate increase of food intake[10], whereas knocking down insulin receptor in the hypothalamus resulted in a more dramatic hyperphagia phenotype[32]. Given that insulin controls neuronal growth and differentiation[47, 48], genetic deletion of insulin signaling throughout the CNS might cause unknown alterations that complicate its function in the hypothalamus. Thus, it is likely that dampening insulin signaling in the hypothalamus, rather than completely abrogating it, could elicit a more significant consequence in food intake activities. Indeed, we found reduced insulin sensitivity in the hypothalamus of MANF transgenic mice. In addition, restoring insulin signaling in the hypothalamus ameliorated the hyperphagia activity in MANF transgenic mice. Altogether, these results support that hypothalamic insulin resistance resulted from increased MANF levels is a major contributor of hyperphagia in MANF transgenic mice. The CNS contains diverse types of anorexigenic and orexigenic neurons, which respond differently to insulin signaling[33, 49, 50]. It remains to be investigated which types of neurons express MANF, and how MANF functions in these neurons to mediate their activities.

It is noteworthy that insulin resistance was only found in tissues with increased MANF expression. For example, in MANF transgenic mice, impaired insulin response was found in the cortex and hypothalamus, both of which express transgenic MANF, but not in the liver, where transgenic MANF is not detected; in WT mice with viral MANF injection, insulin resistance was only found in the hypothalamus, where the virus was delivered. These results suggest it is increased MANF per se, rather than obesity or other secondary effects, that leads to insulin resistance. Interestingly, previous studies found that knockdown of MANF increased, whereas overexpression of MANF inhibited, cell proliferation[45]. Given that insulin is a mitogenic factor, it is possible that MANF also modulates insulin signaling to regulate cell proliferation.

One unique feature of MANF as a non-classical neurotrophic factor is its localization in the ER and function as an ER stress response protein[15]. In this study, we demonstrated that MANF has an additional intracellular function in mediating insulin signaling, and MANF does so by regulating the subcellular localization and function of PIP4k2b. PIP4k2b belongs to the family of type 2 PIP kinases, which catalyzes the conversion of PI-5-P to PI-4,5-$P_2$. Both in vitro and in vivo studies support the notion that PIP4k2b inhibits insulin signaling, possibly by decreasing the level of PI-3,4,5-$P_2$[41, 42, 51]. Moreover, the ER localization appears to be critical for the function of PIP4k2b, as both the enzyme and its substrate are highly enriched in the ER[44, 52]. We found that MANF binds to and regulates the ER localization of PIP4k2b. A mutant form of PIP4k2b with deletion of MANF interacting domain fails to localize in the ER and to inhibit insulin signaling, indicating MANF could play an important role in regulating the enzymatic activity of PIP4k2b. Knocking down PIP4k2b in the hypothalamus of MANF transgenic mice ameliorated, but did not completely eliminate hyperphagia, which could be due to that the remaining activity of PIP4k2b is still sufficient to moderately inhibit insulin response. Alternatively, it is likely that MANF has other unknown mechanisms to modulate food intake activity. Brain injection of recombinant MANF protein has been previously shown neuroprotective in a Parkinson disease rat model[14]. We injected similarly purified recombinant MANF protein into the third ventricle or hypothalamus of WT mice, but did not observe any significant changes in food intake. This result suggests that MANF only functions intracellularly to modulate food intake. However, we cannot rule out the possibility that the recombinant MANF protein purified from bacteria lacks certain post-translational modifications that might be essential for extracellular MANF to regulate food intake.

MANF appears to have multifaceted and cell-type specific functions, as it is recently found to promote activation of innate immune cells during regeneration in the retina[19]. Our study established MANF as an important hypothalamic regulator of feeding behavior, mainly based on the fact that it is upregulated in the hypothalamus by fasting and changing its expression in the hypothalamus can influence feeding activity. Altered function of

**Fig. 7** MANF interacts with PIP4k2b. **a** Recombinant MANF-His protein was mixed with wild type (WT) mouse brain lysate (MANF/lysate). Two control reactions were set: 1. MANF-His protein only (MANF only); 2. WT mouse brain lysate only (Lysate only). The reactions were incubated at 4 °C overnight, and then Ni-NTA beads were added to pull down MANF-His protein together with its interacting partners. **b** A single image of silver stain result of His-pull down in Fig. 7a. The band that is specifically present in MANF/lysate is indicated by a *black arrow*. **c** Representative western blotting images (from three independent experiments) showed that GST-PIP4k2b was able to precipitate MANF-His. **d** Co-immunoprecipitation study using PC12 cells transfected with MANF-HA and Myc-PIP4k2b. Representative western blotting images (from three independent experiments) showed that Myc-PIP4k2b was detected in the pull-down lysate. **e** Co-immunoprecipitation study using MANF transgenic mouse brain lysate. Representative western blotting images (from three independent experiments) showed that PIP4k2b was detected in the pull-down lysate. **f** Three different GST tagged PIP4k2b fragments (Δ1, Δ2 and Δ3) were generated as demonstrated in the figure. Representative western blotting images (from three independent experiments) showed that fragment Δ1 was able to precipitate MANF-His. **g** Representative western blotting images (from three independent experiments) of subcellular fractionation using wild type (WT) and MANF transgenic (TG) mouse brain lysate. Fractions of nucleus (nuc), cytosol (cyto), endoplasmic reticulum (ER) and mitochondria (mito) were indicated by antibodies against LSD1, MEK, PDI and COX IV respectively. **h** Representative western blotting images (from three independent experiments) of PIP4k2b levels in either total or ER fraction of WT and TG mice. **i** Quantification of western blotting results in Fig. 7h (*$P < 0.05$, $n = 3$; 2-tailed student $t$ test, Brain, $t = 2.99$, $P = 0.0480$; Liver, $t = 1.376$, $P = 0.3026$). Data are represented as mean ± SEM

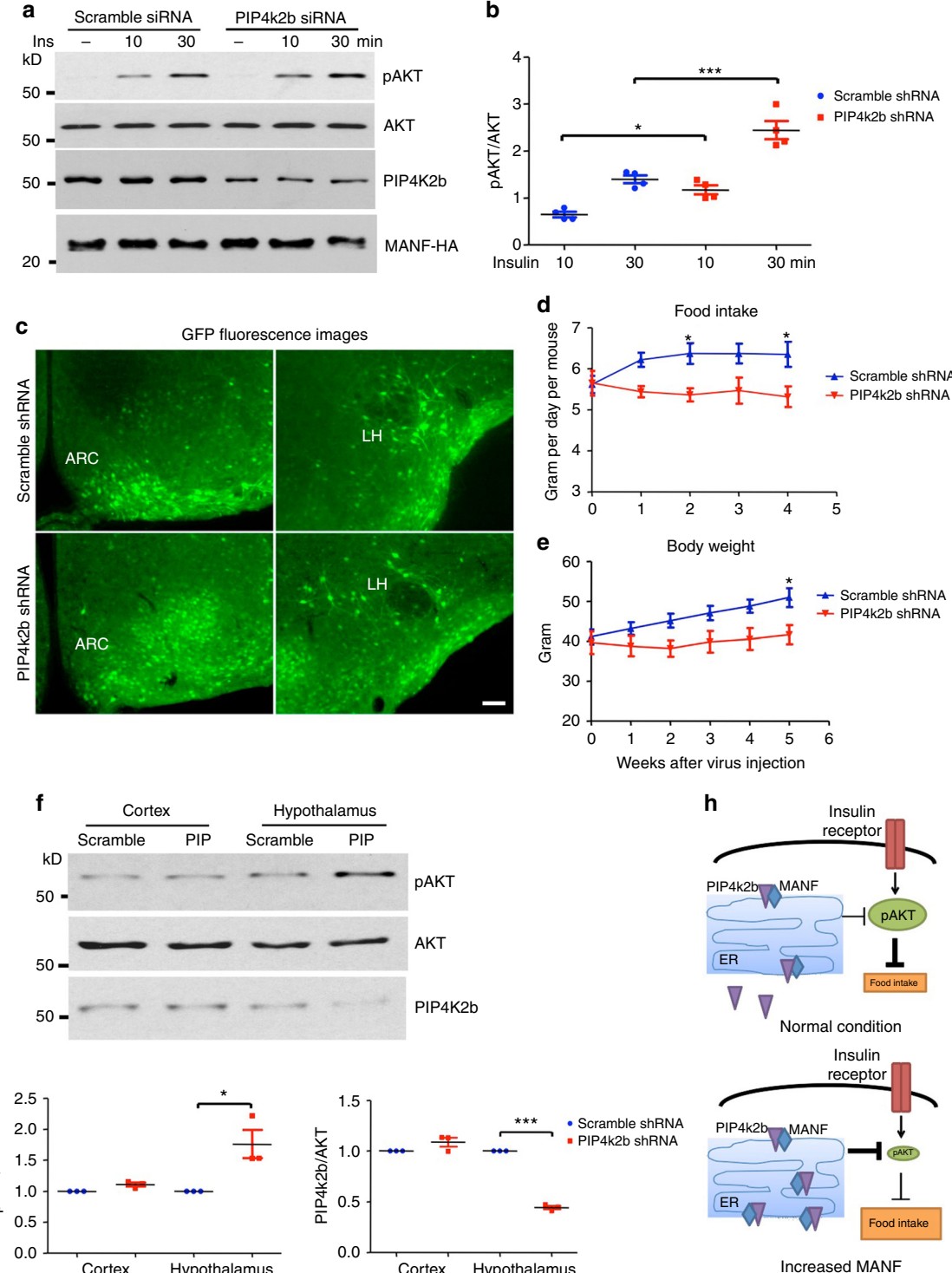

**Fig. 8** Reducing PIP4k2b ameliorates MANF mediated insulin resistance. **a** Representative western blotting images (from three independent experiments) of pAKT levels in PC12 cells treated with insulin (Ins) for different lengths of time: untreated (−), 10 min or 30 min. **b** Quantification of western blotting results in Fig. 8a (*P < 0.05, ***P < 0.001, n = 4, one-way ANOVA with Tukey post-tests, F = 39.17, P < 0.0001). **c** GFP fluorescence was used to indicate the areas with shRNA expression in mice injected with either AAV-scramble-shRNA or AAV-PIP4k2b-shRNA (*Scale bar*: 100 μm). **d** Food intake of MANF TG mice injected with either AAV-scramble-shRNA or AAV-PIP4k2b-shRNA was measured weekly after viral injection (*P < 0.05, n = 6, two-way ANOVA with Bonferroni post-tests, F = 23.17, P < 0.0001). **e** Body weights of MANF TG mice injected with either AAV-scramble-shRNA or AAV-PIP4k2b-shRNA were measured weekly after viral injection (*P < 0.05, n = 6, two-way ANOVA with Bonferroni post-tests, F = 23.63, P < 0.0001). **f** Representative western blotting images (from three independent experiments) of pAKT levels in MANF TG mice injected with either AAV-scramble-shRNA (Scramble) or AAV-PIP4k2b-shRNA (PIP) in the hypothalamus, and then treated with insulin. **g** Quantification of western blotting results in Fig. 8f (*P < 0.05, ***P < 0.001, n = 3; 2-tailed student t test, pAKT, t = 3.323, P = 0.0399; PIP4k2b, t = 33.09, P = 0.0009). Data are represented as mean ± SEM. **h** The proposed model of the function of MANF in mediating insulin signaling and food intake

MANF could have a significant impact on energy homeostasis, which potentially leads to obesity and other metabolic disorders. In support of this idea, a recent study identified MANF as a candidate disease gene in a patient with type 2 diabetes mellitus and obesity[53]. Unlike the classic type of neurotrophic factors or hormones that act on the plasma member receptors to produce their intracellular effects on hypothalamic neurons, MANF regulates hypothalamic intracellular signaling by its association with interacting proteins in ER. Thus, our findings provide insight into an additional intracellular mechanism for hypothalamic control of food intake and energy metabolism.

## Methods

**Mouse lines and behavior tests**. All the mice (C57BL/6, Jackson Laboratory) were bred and maintained in the animal facility at Emory University under specific pathogen-free conditions. All procedures were performed in accordance with the NIH and U.S. Public Health Service's Guide for the Care and Use of Laboratory Animals, and were approved by the Institutional Animal Care and Use Committee at Emory University, which is accredited by the American Association for Accreditation of Laboratory Care (AAALC). The mice (5 per cage maximum) were housed in a 12-h light (0700 h–1900 h) and 12-h dark (0700 h–1900 h) cycle at a controlled temperature (22 °C), and fed ad libitum a regular chow diet (LabDiet, 5001, metabolizable energy 2.91 kcal gm$^{-1}$). MANF transgenic mouse model was generated in the previous study[17].

For behavioral tests, mice were randomly assigned to the experimental groups, and the tests were performed by persons who did not know the conditions of each experimental group. To measure food and water intake, both male and female WT and MANF transgenic mice were individually housed in cages. The amount of food and water in each cage was measured at 1400 h on day one, and re-measured at 1400 h on day two. Daily food and water consumption was calculated by subtracting the amount on day two from the amount on day one. The measurement was performed monthly from 1-month to 4-month of age. Locomotor test was performed as described previously[54]. Briefly, 2-month old male and female WT and MANF transgenic mice were individually placed in the chambers from an automated system (San Diego Instruments), with free access to food and water. The mice were allowed to acclimate to the environment for 6 h, and their activities were recorded every 30 min during a 12-h light and 12-h dark cycle. Indirect calorimetry of 3-month old male and female WT and MANF transgenic mice was performed at Vanderbilt Mouse Metabolic Phenotyping Center. The Energy Expenditure ANCOVA analysis done for this work was provided by the NIDDK Mouse Metabolic Phenotyping Centers (MMPC, www.mmpc.org) using their Energy Expenditure Analysis page (http://www.mmpc.org/shared/regression.aspx) and supported by grant DK076169.

**Antibodies and plasmids**. Primary antibodies used in this study include: MANF (LSBio, C53208, B2688; Abcam, ab67271; 1:1000), Actin (Sigma, A5060; 1:10,000), HA (Cell signaling, 2367, 3724; Roche, 12CA5; 1:10,000), Vinculin (Sigma, V9131; 1:10,000), pAKT (Cell signaling, 4051, 4060; 1:1,000), AKT (Cell signaling, 9272; 1:1000), His (Santa Cruz, 803; 1:1000), GST (Sigma, G7781; 1:1000), Myc (Cell signaling, 2272; 1:1000), PIP4k2b (Proteintech, 13218-1-AP; 1:500), LSD1 (Cell signaling, 2139; 1:1000), MEK (Cell signaling, 4694; 1:1000), PDI (Cell signaling, 2446; 1:500), COX IV (Cell signaling, 4844; 1:1000), pIKK (Cell signaling, 2078; 1:500), pJNK (Santa Cruz, 6254; 1:1000), POMC (Santa Cruz, 20148; 1:200), IRβ (Santa Cruz, 711; 1:1000), pSTAT3 (Cell signaling, 9145; 1:1000), STAT3 (Cell signaling, 9139; 1:1000).

To generate MANF plasmid, MANF cDNA was amplified from mouse brain cDNA library by PCR with primers (forward: 5′-ATG GAT CCA GGA TGT GGG CTA CGC-3′; reverse: 5′-ATG AAT TCC AGA TCA GTC CGT GCG-3′), and inserted into pRK5 vector at BamHI and EcoRI restriction sites. The sequence encoding HA tag has been previously engineered into the pRK5 vector so that it is linked to the C-terminus of MANF. To generate PIP4k2b plasmid, PIP4k2b cDNA was amplified from mouse brain cDNA library by PCR with primers (forward: 5′-TAG GAT CCG CCA CCA TGT CGT CCA ACT GCA CCA G-3′; reverse: 5′-TAT CTA GAT CAC AGA TCT TCT TCA GAA ATA AGT TTT TGT TCC GTC AGG ATG TTG GAC ATG AAC-3′), and inserted into pRK5 vector at BamHI and XbaI restriction sites. The sequence encoding Myc tag was added to the reverse primer. To generate truncated PIP4k2b plasmid, full length PIP4k2b plasmid was used as template for PCR with primers (forward: 5′-TAG GAT CCG CCA CCA TGA GGA GGT TCG GCA TCG ATG-3′; the same reverse primer), and inserted into pRK5 vector at BamHI and XbaI restriction sites.

**Virus generation and stereotaxic injection**. To generate AAV-MANF viral vector, the sequence encoding MANF with HA tag was released from pRK5-MANF plasmid by restriction digestion, and inserted into AAV-UBC viral vector by blunt end ligation. AAV-PIP4k2b-shRNA viral vector (target sequence: 5′-GGG TGA ACC ACA CGA TCA ATG-3′) was purchased from Vector Biolabs. AAV-MANF shRNA viral vector was generated by replacing PIP4k2b shRNA with MANF

shRNA (target sequence: 5′-CCG TGA AGC AAG AGG CAA A-3′). PX551 (AAV-Mecp2-Cas9) and PX552 were gifts from Feng Zhang at the Massachusetts Institute of Technology (Addgene plasmid #60957, 60958). AAV-MANF-sgRNA viral vector was generated by inserting MANF sgRNA (target sequence: 5′-GGA GGA TGT GGG CTA CGC GC-3′) into PX552. AAV-Gfap-Cas9 was generated by replacing Mecp2 promoter in PX551 with Gfap promoter. These viral vectors were sent to the Viral Vector Core at Emory University for packaging (AAV9 serotype, 1.5 × 10$^{13}$ vg ml$^{-1}$). Pooled lentivirus-PIP4k2b-siRNA and lentivirus-scramble-siRNA were purchased from Applied Biological Materials. AAV-scramble-shRNA was purchased from Vector Biolabs.

Method for stereotaxic viral injectionwas adopted from our previous study[55]. The mice were anesthetized with 1.5% isoflurane inhalation, and stabilized in a stereotaxic instrument (David Kopf Instruments). The location of the injected site was determined according to the distance from bregma: anteriorposterior = −1.5 mm, medial-lateral = ± 0.35 mm, dorsal-ventral = - 5.5 mm. Small holes were drilled on the skull, and a 30-gauge Hamilton microsyringe was used to deliver the virus at a speed of 200 nl per min (bilateral injection, 1 μl in each side). For the ARC injection (bilateral injection, 0.2 μl in each side, 50 nl per minute), the coordinates are anteriorposterior = −1.5 mm, medial-lateral = ± 0.2 mm, dorsal-ventral = −5.8 mm; for the LH injection (bilateral injection, 0.2 μl in each side, 50 nl per minute), the coordinates are anteriorposterior = −1.5 mm, medial-lateral = ± 1 mm, dorsal-ventral = −5.1 mm. For injection into the third ventricle, the coordination used was anteriorposterior = −1.5 mm from bregma and on the midline, dorsal-ventral = −5 mm. Meloxicam (2 mg kg$^{-1}$) was given as analgesics, and after surgery the mice were placed on a heated blanket to recover from anesthetic.

**Immunoprecipitation and in vitro binding**. Method for immunoprecipitation was described previously[56, 57]. Briefly, cell or brain lysates were homogenized in NP-40 buffer (50 mM NaCl, 50 mM Tris–HCl pH8.0, 0.1% Triton X-100, 0.5% NP-40), and 300 μg of protein was used for one experiment. The protein was first pre-cleared with protein A-agarose for 1 h, and then incubated with primary antibody overnight. The next day, protein A-agarose was added into the mixture and kept for 1 h incubation. The precipitated antibody-protein complexes were collected.

For in vitro binding assay, MANF-His protein was mixed with GST or GST-PIP4k2b beads in 500 μl of NP40 buffer, and kept at 4 °C overnight. On the next day, the beads were washed with NP40 buffer three times, and the precipitated beads–protein complexes were used for western blotting.

To perform affinity purification mass spectrometry, MANF-His protein was mixed with 500 μg of mouse brain lysate in 500 μl of NP40 buffer. After overnight incubation at 4 °C, Ni-NTA beads were added to the mixture and kept for 1 h. The beads were washed with NP40 buffer three times, and the precipitated beads–protein complexes were loaded to a Tris-Glycine gel (ThermoFisher Scientific) for electrophoresis. Silver staining was performed using a commercial kit (Bio-Rad). The identified protein bands were excised from the gel and sent to Emory Integrated Proteomics Core for mass spectrometry analysis.

**Blood glucose measurement and plasma leptin ELISA**. To measure blood glucose, the mice were fasted overnight with free access to water. The blood was drawn by either tail clipping. Blood glucose was measured with a glucose meter (Roche, Accu-Chek). For glucose tolerance test, the mice were fasted overnight, and given an intraperitoneal injection of glucose at 2 g kg$^{-1}$ body weight. For insulin tolerance test, the mice were fasted for 5 h, and given an intraperitoneal injection of insulin at 0.75 U kg$^{-1}$ body weight. Plasma leptin was measured with a mouse leptin ELISA kit (Crystal Chem), according to the kit manual.

**Cell culture**. Mycoplasma-free PC12 cells (ATCC, CRL-1721) were cultured in Dulbecco's modified Eagle's medium (DMEM) supplemented with 10% horse serum, 5% fetal bovine serum and 100 U ml$^{-1}$ penicillin. Lipofectamine 2000 (ThermoFisher Scientific) was used for transfection. For insulin treatment, the cells were serum-starved in DMEM for 4 h, and then insulin (Sigma, I9278) was added to DMEM.

**Recombinant protein production**. His tagged MANF protein production was described previously[17]. Mouse MANF cDNA was cloned into pET-28a vector and transformed into XL1-Blue competent cells. MANF production was induced by incubating the cells with IPTG for 1 h at 37 °C. The cells were lysed in lysis buffer (5 mM imidazole, 500 mM NaCl, 20 mM Tris–HCl, pH 8.0, 20 mM Beta-mercaptoethanol, 1 mM PMSF) by sonication, and incubated with Ni-NTA beads (QIAGEN) at 4 °C for 2 h. The beads with lysate was loaded to Poly-Prep Chromatography column (BIO-RAD), and washed with washing buffer (15 mM imidazole, 500 mM NaCl, 20 mM Tris–HCl, pH 8.0, 20 mM Betamercaptoethanol, 0.1% NP40, 1 mM PMSF) three times. MANF protein was eluted by elution buffer (400 mM imidazole, 500 mM NaCl, 20 mM Tris–HCl, pH 8.0, 20 mM Betamercaptoethanol, 0.1% NP40, 1 mM PMSF), and concentrated using Amico Ultra-4 centrifugal filters (Millipore). MANF protein aliquots, dissolved in PBS, were kept at −80 °C.

To generate GST tagged full length PIP4k2b and PIP4k2b fragments, mouse PIP4k2b cDNA was cloned into pGEX-4T-1 vector. Primers used are listed below: full length PIP4k2b (forward: 5′-TAG GAT CCG CCA CCA TGT CGT CCA ACT

GCA CCA G-3′; reverse: 5′-CCG CTC GAG TCA CGT CAG GAT GTT GGA CAT GAA C-3′); fragment Δ1 (forward: 5′-TAG GAT CCG CCA CCA TGT CGT CCA ACT GCA CCA G-3′; reverse: 5′-CCG CTC GAG TCA AGT GGA ACC CTT CAG GTC C-3′); fragment Δ2 (forward: 5′-TAG GAT CCC CTA TCA ACA GTG ACA GCC A-3′; reverse: 5′-CCG CTC GAG TCA GTG GAT GCC CAC CAG GAG AC-3′); fragment Δ3 (forward: 5′-TAG GAT CCC ATC GGC TTA CTG TGC ATC G-3′; reverse: 5′-CCG CTC GAG TCA CGT CAG GAT GTT GGA CAT GAA C-3′). Those vectors were transformed into DH5α competent cells, and protein production was induced by incubating the cells with IPTG for 1 h at 37 °C. The cells were lysed in lysis buffer (1% Triton X-100 and 1:500 PMSF in 1× PBS) by sonication, and then mixed with glutathione beads (Sigma) at 4 °C overnight. The beads were washed (0.5% Triton X-100 and 1:500 PMSF in 1× PBS) three times, re-suspended in 1× PBS and stored at 4 °C.

**RNA extraction and real time PCR**. Methods used for quantitative real time PCR were described previously[17]. RNA from mouse hypothalamus was isolated by RNeasy Lipid Tissue Mini Kit (QIAGEN, 74804). Equal amount of RNA was used for cDNA synthesis with SuperScript III First-Strand synthesis system (Invitrogen, 18080-051). For real time PCR, RealMasterMix 2.5× from 5 Prime was used. The PCR reaction was performed in Mastercycler realplex (Eppendorf). Primers used are listed below. For MANF, forward: 5′-ATT GAC CTG AGC ACA GTG GAC CTG-3′; reverse: 5′-TTC AGC ACA GCC TTT GCA CAT CTC-3′. For Actin, forward: 5′-TCA CTG TCC ACC TTC CAG CAG ATG-3′; reverse: 5′-CTC AGT AAC AGT CCG CCT AGA AGC-3′.

**Subcellular fractionation**. The method for subcellular fractionation was adopted from a previous publication[58]. Brains or cells were homogenized by a glass grinder in buffer containing 0.25 M sucrose, 10 mM HEPES, pH 7.5. The homogenate was centrifuged at 700 g for 10 min. The pellet (P1) was re-suspended in 1.8 M sucrose and centrifuged at 70,900×g for 90 min to get the pellet (P2) as nuclei fraction. The supernatant (S1) was centrifuged at 6000×g for 15 min to get the pellet (P3) as mitochondria fraction. The supernatant (S2) was centrifuged at 124,000×g for 60 min to get the pellet (P4) as ER fraction, and the supernatant (S3) was considered as the cytosol.

**Western blotting and immunohistochemistry**. For western blotting, mouse brain tissues or collected cells were lysed in ice-cold RIPA buffer containing Halt protease inhibitor cocktail and phosphatase inhibitors. The lysates were incubated on a rocker at 4 °C for 30 min, sonicated, and centrifuged at top speed for 10 min. The supernatants were collected and subjected to SDS–PAGE. The proteins on the gel were transferred to a nitrocellulose membrane. The blot was blocked with 5% milk/PBS for 1 h at room temperature and incubated with a primary antibody in 3% BSA/PBS overnight at 4 °C. After three washes in PBS, the blot was incubated with HRP-conjugated secondary antibodies in 5% milk/PBS for 1 h at room temperature. The blot was then washed three times in PBS, and developed using ECL Prime (GE Healthcare).

For immunostaining, fixed brain sections or cultured cells were blocked with 3% bovine serum albumin in PBS supplemented with 0.2% Triton X-100 for 30 min at room temperature. Slices were incubated with primary antibodies at 4 °C overnight and washed with PBS three times, secondary antibodies and nuclear dye Hoechst were added to the samples for 1 h at room temperature. Images were acquired with an Imager Z1 microscope equipped with objective lens. The representative images were selected from at least 5–6 images that were used for analysis.

**Statistical analysis**. For mouse behavioral analysis, each group consisted of at least five animals. For western blotting analysis, immunostaining, or other biochemical assays, data were generated from three or more experiments, and the results were expressed as mean ± SEM. Statistical significances were calculated based on either two-tailed student $t$ test, one-way ANOVA or two-way ANOVA. A $P$-value <0.05 was considered significant.

**Data availability**. The data that support the findings of this study are available from the corresponding authors upon reasonable request.

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

## Acknowledgements

The work was supported by NIH grants (NS102913-01, NS101701-01 to X.-J.L. and NS095279 to S.L.) and the National Natural Science Foundation of China (91332206, 81461148020). This research project was supported in part by the Viral Vector Core and the Emory University Integrated Proteomics Core of the Emory Neuroscience NINDS Core Facilities grant, P30NS055077. We thank Vanderbilt Mouse Metabolic Phenotyping Center (DK059637) for performing indirect calorimetry, Dr. Feng Zhang for his contribution of CRISPR related plasmids, and Grace Wynn for her help with mouse behavioral tests.

## Author contributions

S.Y. designed and performed experiments, analyzed data and wrote the paper. H.Y., R.C., P.Y., Y.Y., W.Y. and S.H. performed experiments. M.A.G. assisted with mouse breeding. S.L. and X.-J.L. designed experiments, analyzed data and wrote and edited the paper.

## Additional information

**Competing interests:** The authors declare no competing financial interests.

