## [Peer Review File · Nature Communications]

Reviewers' comments:

Reviewer #1 (Remarks to the Author):

The manuscript by Su Yang et al showed that MANF expression was increased in the hypothalamus of mice after fasting. In MANF transgenic mice and the wild type mice treated with viral MANF in the hypothalamus, MANF overexpression led to hyperphagia and obesity. On the contrary, MANF knockdown in hypothalamus led to hypophagia and retarded body weight gain. The author also claimed that MANF interacted with PIP4k2b in the ER and recruited PIP4k2b to localize in ER to mediate insulin resistance. The author also showed PIP4k2b knockdown in the hypothalamus improved hypothalamic insulin sensitivity and ameliorates hyperphagia in MANF transgenic mice. These results indicate MANF is involved in hypothalamus-controlled food intake and energy homeostasis by interacting PIP4k2b. This study was well planned and most of the experimental data have positive results. However, there are a few weaknesses. Many issues need to be addressed.

Major points:

1. In Figure 1, PIP4k2b level and insulin receptor should be detected in cortex and hypothalamus of mice after 48 h fasting.
2. Why was MANF not detectable in liver in MANF TG mice?
3. In Figure 6a, the levels of pAKT were increased in cortex and hypothalamus in MANF TG mice without insulin (lane 3 vs lane1), which is more obvious than that after treatment with insulin. Does it mean that TG can induce AKT activation?
4. In Figure 6b, why are there many bands in MANF only lane? The gel should be transferred and blotted with anti- PIP4k2b antibody.
5. In MANF TG mice, PIP4k2b was upregulated in the ER of brain tissue (Figure 7g). How about PIP4k2b level in MANF TG mice after treated with insulin? PIP4k2b levels in the hypothalamus neurons in the MANF overexpression or knockdown mice should be investigated.
6. The level of PIP4k2b in the ER fraction was greatly increased in the brain lysate of MANF transgenic mice (In Figure 7g), suggesting MANF can upregulate PIP4k2b expression. The authors should discuss how MANF upregulates PIP4k2b expression. Furthermore, PIP4k2b was specifically upregulated in the ER, but not in cytoplasm, in MANF TG mice, and the total level of PIP4k2b did not change. As we know the newly synthesized proteins need to be modified and folded in ER during mature. It's easy to understand that the overexpressed proteins are enriched in ER. The authors did not mention how to isolate cytoplasm. I want to know if the cytoplasm includes the small organelles, such as ER. IF the cytoplasm did not exclude ER, how to explain PIP4k2b was increased in ER, but not in cytoplasm? If PIP4k2b was only increased in ER, and if MANF can recruit PIP4k2b into the ER, how PIP4k2b regulates the cytoplasmic AKT and its downstream insulin pathway (Fig. 8h).
7. Does PIP4k2b affect MANF expression ?
8. In Figure 8f, a decreased level of PIP4k2b was only found in the hypothalamus of mice injected with AAV-PIP4k2b-shRNA. Why PIP4k2b was not decreased in cortex with the injection of AAV-PIP4k2b-shRNA? Is it a technical problem? How about MANF expression in the hypothalamus of TG mice injected with AAV-PIP4k2b-shRNA?
9. In the hypothalamus of TG mice, how about the expression of insulin receptor?

Minor points:

1. In figure legends, the authors described the each panel as Figure plus lowercase letter. I think the number should be included. For example, the panel b in Figure 1 should refer to Figure 1b or panel b, but not Figure b.
2. In Figure 2a, why the values of three samples are almost the same?

Reviewer #2 (Remarks to the Author):

The manuscript by Yang and colleagues investigates the physiological roles of mesencephalic astrocyte-derived neurotrophic factor (MANF) in the control of energy intake. The authors

demonstrate that overexpression of MANF in the mouse brain induces hyperphagia and obesity. The authors also describe that this increased body weight is due to MANF-mediated insulin resistance in the hypothalamus. They suggest that increased MANF enhances PIP4k2b activity in the ER. Based on these findings, the authors conclude that hypothalamic MANF plays a critical role in the regulation of overall energy balance. Although this is a nice study, my enthusiasm for this study is dampened by some major issues described below. In addition, the cellular mechanisms underlying the orexigenic effect of MANF described in the manuscript are not consistent with the prior studies. In fact, MANF is upregulated by ER stress and protects cells against ER stress-induced cell death as described in the introduction. Moreover, hypothalamic ER stress appears to be a causative factor for the development of leptin resistance and consequently obesity (Schneeberger et al. 2013, Cell). In other words, increased MANF may counteract ER stress-induced obesity.

Major comments:

1. Figure 1 shows expression of MANF under physiological conditions. However, results are not consistent with what they describe in the text. In fact, it looks like that expression of MANF is much higher in the hypothalamus than in the cortex (1a). However, western blot analysis shows the opposite (1b). In both cases (fed and fasted), the expression of MANF protein appears to be higher in the cortex than in the hypothalamus (1b). In addition, levels of MANF expression in the ARC and LH after fasting (1d) are very similar to those shown in Fig. 1a. Hence, it is unclear whether the hypothalamus is a major area that expresses MANF and whether the expression of MANF is actually regulated by the availability of nutrients.
2. Figure 2 and 3 show effects of transgenic MANF expression in the brain. Transgenic mice show hyperphagia and obesity. Importantly, transgenic animals that are pair-fed to WT littermates show no increase in body weight, suggesting that altered food intake results in obesity in transgenic mice. Although data are nice, one needs to know if injection of MANF into the hypothalamus can also promote feeding.
3. Figure 4 describes effects of MANF overexpression in the hypothalamus. According to their description, AAV vectors were injected into the ventromedial hypothalamus (AP = - 1.5 mm, ML = \pm 0.35 mm, DV = - 5.5 mm). If this is the case, MANF expression should be restricted in the area where viruses are injected such as the VMH but not throughout the hypothalamus. The authors do not provide detailed information as to their AAV vectors (titer, serotype, volume, etc). This issue appears to be important to understand how MANF regulates food intake and body weight. In fact, the hypothalamus contains diverse types of neurons such as anorexigenic and orexigenic cells. Since there are NPY, POMC, TH, orexin, and MCH neurons in the ARC and LH, one needs to know what neurons in the ARC, LH and DMH actually express MANF. Based on those results, they can inject viruses specifically into the ARC or the LH.
4. In page 10 (line 217-220), the authors describe a reduction of POMC expression in mice injected with AAV-MANF due to insulin resistance. Interestingly, the study of Konner and colleagues (2007) shows that insulin signaling in POMC neurons is not involved in feeding behavior. Although all the data described in the manuscript suggest the orexigenic effect of MANF, the cellular mechanism(s), by which MANF promotes feeding are not discussed in the text.
5. The authors conclude that increased food intake and body weight in mice overexpressing MANF in the hypothalamus are mediated by insulin resistance. It is true that AKT plays an important role in the insulin-signaling pathways. However, it is not clear if direct injection of insulin into the hypothalamus of mice overexpressing MANF in the hypothalamus has no effect on food intake due to insulin resistance.
6. Finally, the cellular mechanisms underlying the orexigenic effects of MANF described in the manuscript are not consistent with the prior studies. In fact, MANF is upregulated by ER stress and protects cells against ER stress-induced cell death. Moreover, hypothalamic ER stress appears to be a causative factor for the development of leptin resistance and consequently obesity (Schneeberger et al. 2013, Cell). In other words, increased MANF may rather inhibit leptin and insulin resistance.

Minor comments:

None

Reviewer #3 (Remarks to the Author):

Yang et al reported interesting findings that hypothalamic MANF is previously unknown regulator of energy balance. Using various mouse models, the authors convincingly demonstrated that manipulating hypothalamic MANF profoundly affects food intake and body weight. Mechanistically, the authors proposed that MANF suppresses hypothalamic insulin action, which explains the observed phenotypes. However, brain specific knockout of insulin receptors has modest effects on food intake and body weight. Moreover, there is no data to exclude the possibility that secreted MANF may also promote weight gain independently of insulin. The study is interesting and potentially important, and could be improved by additional experiments to clarify confusions with regard to the underlying mechanisms.

1. It is important to determine the contribution of secreted MANF to energy balance and weight control. Does icv injection of recombinant MANF (the secreted forms) affect food intake and body weight? Does overexpression of the secreted portion of MANF in the hypothalamus affect energy balance and body weight?
2. It is premature to conclude that MANF regulates food intake via suppressing hypothalamic insulin signaling. Does MANF affect leptin signaling? Plasma leptin levels and hypothalamic leptin signaling should be examined in MANF-expressing and knockdown mice before the onset of obesity.
3. Do hypothalamic expression and ER localization of PIP4k2b and MANF correlate with obesity development?
4. Hypothalamic insulin is known to regulate systemic glucose metabolism and liver lipid metabolism. Plasma glucose levels, GTT, ITT, and liver steatosis should be assessed in MANF overexpressing or silencing mice.

Reviewer #1

The manuscript by Su Yang et al showed that MANF expression was increased in the hypothalamus of mice after fasting. In MANF transgenic mice and the wild type mice treated with viral MANF in the hypothalamus, MANF overexpression led to hyperphagia and obesity. On the contrary, MANF knockdown in hypothalamus led to hypophagia and retarded body weight gain. The author also claimed that MANF interacted with PIP4k2b in the ER and recruited PIP4k2b to localize in ER to mediate insulin resistance. The author also showed PIP4k2b knockdown in the hypothalamus improved hypothalamic insulin sensitivity and ameliorates hyperphagia in MANF transgenic mice. These results indicate MANF is involved in hypothalamus-controlled food intake and energy homeostasis by interacting PIP4k2b. This study was well planned and most of the experimental data have positive results. However, there are a few weaknesses. Many issues need to be addressed.

Major points:

1. In Figure 1, PIP4k2b level and insulin receptor should be detected in cortex and hypothalamus of mice after 48 h fasting.

We checked the expression of PIP4k2b and Insulin receptor (IR β) in the cortex and hypothalamus of mice after 48 h fasting, and did not find any significant difference when compared with *ad libitum* fed mice. We included this negative result as **new Supplementary Fig. 8a**.

2. Why was MANF not detectable in liver in MANF TG mice?

In **Fig. 2c**, we used HA antibody to detect transgenic MANF (tagged with HA) expression. In our transgenic mouse model, the expression of transgenic MANF is driven by the mouse prion promoter, which is a neuronal promoter that drives gene expression mainly in neuronal cells^{1,2}. Therefore, it is expected that transgenic MANF was not detectable in the liver.

3. In Figure 6a, the levels of pAKT were increased in cortex and hypothalamus in MANF TG mice without insulin (lane 3 vs lane1), which is more obvious than that after treatment with insulin. Does it mean that TG can induce AKT activation?

This is an interesting question. Because MANF functions both intracellularly and extracellularly, it is possible that extracellular MANF could induce AKT activation. However,

the signaling pathways mediated by extracellular MANF remains poorly understood and needs to be investigated by future studies.

In the current study, we focused on the insulin signaling. In **Fig. 6a**, although the basal level of pAKT was higher in MANF TG mice, administration of insulin failed to increase pAKT level, indicating that MANF TG mice are insulin resistant. In the revised manuscript, we also performed additional experiments, and the results also support insulin resistance in MANF TG mice (**new Fig. 6e-g, Supplementary Fig. 6a**).

4. In Figure 6b (it should be 7b), why are there many bands in MANF only lane? The gel should be transferred and blotted with anti- PIP4k2b antibody.

The recombinant MANF protein (with His tag) used in **Fig. 7b** was purified from bacteria using Ni-NTA agarose. Although MANF is the most abundant protein after purification, it is difficult to avoid contamination by some bacterial proteins, which were revealed by the sensitive silver staining method. Nonetheless, we only cut the band specifically found in the MANF/lysate lane for mass spectrometry analysis, and the proteins in the band were identified by sequence matching to a mouse proteomic database, so that these bacterial proteins could not interfere with the mass spectrometry result.

5. In MANF TG mice, PIP4k2b was upregulated in the ER of brain tissue (Figure 7g). How about PIP4k2b level in MANF TG mice after treated with insulin? PIP4k2b levels in the hypothalamus neurons in the MANF overexpression or knockdown mice should be investigated.

We examined PIP4k2b expression in MANF TG mice after insulin treatment and in the hypothalamus of MANF knockdown mice. We did not find any significant differences (**new Supplementary Fig. 8b, c**). This result is consistent with our original finding that MANF overexpression in the transgenic mice did not change the expression of PIP4k2b, but altered its subcellular localization to be more enriched in the ER (**Fig. 7g, h**).

6. The level of PIP4k2b in the ER fraction was greatly increased in the brain lysate of MANF transgenic mice (In Figure 7g), suggesting MANF can upregulate PIP4k2b expression. The authors should discuss how MANF upregulates PIP4k2b expression. Furthermore, PIP4k2b was specifically upregulated in the ER, but not in cytoplasm, in MANF TG mice, and the total level of PIP4k2b did not change. As we know the newly synthesized proteins need to be modified and folded in ER during mature. It's easy to

understand that the overexpressed proteins are enriched in ER. The authors did not mention how to isolate cytoplasm. I want to know if the cytoplasm includes the small organelles, such as ER. IF the cytoplasm did not exclude ER, how to explain PIP4k2b was increased in ER, but not in cytoplasm? If PIP4k2b was only increased in ER, and if MANF can recruit PIP4k2b into the ER, how PIP4k2b regulates the cytoplasmic AKT and its downstream insulin pathway (Fig. 8h).

According to our original (**Fig. 7h**) and additional (**new Supplementary Fig. 8**) results, changes in MANF expression did not affect the overall level of PIP4k2b in the cells. Instead, MANF could alter the subcellular localization of PIP4k2b, and overexpression of MANF recruits more PIP4k2b into the ER.

We used a series of centrifuging steps to separate each subcellular compartment. In our purification protocol, after centrifuging at 124000 g for 60 minutes, the pellet was the ER fraction, and the supernatant was the cytoplasm. Therefore the cytoplasm did not contain ER, which was demonstrated in **Fig. 7g**, as the cytoplasm had minimal staining for the ER marker PDI. We have added more details of our fractionation protocol in the Supplemental Methods to avoid any confusion.

According to previous literatures, the enzymatically active PIP4k2b was found in the ER³⁻⁵, where it catalyzes the conversion of PI-5-P to PI-4,5-P₂. The product could diffuse into the cytoplasm and regulate AKT activity and the downstream insulin pathway. Both *in vitro* and *in vivo* studies demonstrated that PIP4k2b inhibits insulin signaling^{3,6,7}. We included the above information in the Discussion.

7. Does PIP4k2b affect MANF expression?

We examined MANF expression in MANF transgenic mice injected with viral PIP4k2b shRNA, and did not find any significant differences (**new Supplementary Fig. 10**).

8. In Figure 8f, a decreased level of PIP4k2b was only found in the hypothalamus of mice injected with AAV-PIP4k2b-shRNA. Why PIP4k2b was not decreased in cortex with the injection of AAV-PIP4k2b-shRNA? Is it a technical problem? How about MANF expression in the hypothalamus of TG mice injected with AAV-PIP4k2b-shRNA?

We injected AAV-PIP4k2b-shRNA into the hypothalamus of MANF TG mice, so that PIP4k2b expression was only decreased in the hypothalamus, but not in the cortex. We included the cortical samples as a control in **Fig. 8f** to demonstrate that the increased AKT phosphorylation was specifically found in the hypothalamus with PIP4k2b knockdown, but

not in the cortex. To avoid possible confusions, we revised the text and figure legend for **Fig. 8f** to clearly indicate that AAV-PIP4k2b-shRNA was delivered to the hypothalamus. We examined MANF expression in MANF TG mice injected with AAV-PIP4k2b-shRNA, and did not find any significant differences (**new Supplementary Fig. 10**).

9. In the hypothalamus of TG mice, how about the expression of insulin receptor?

We checked the expression of insulin receptor (IR β) in the hypothalamus of wild type and MANF TG mice, and did not find any significant difference (**new Supplementary Fig. 6d**).

Minor points:

1. In figure legends, the authors described the each panel as Figure plus lowercase letter. I think the number should be included. For example, the panel b in Figure 1 should refer to Figure 1b or panel b, but not Figure b.

We corrected this issue in the revised text.

2. In Figure 2a, why the values of three samples are almost the same?

We calculated the relative abundance of MANF mRNA in wild type and MANF TG mice using the double delta Ct method ($2^{-\Delta\Delta CT}$). This method assumes the mRNA level in one sample (in our case, the wild type mice) as one, and then calculates the relative fold change of mRNA level in the other sample (in our case, the MANF TG mice). Therefore, the values for all 3 WT mice are one.

Reviewer #2

The manuscript by Yang and colleagues investigates the physiological roles of mesencephalic astrocyte-derived neurotrophic factor (MANF) in the control of energy intake. The authors demonstrate that overexpression of MANF in the mouse brain induces hyperphagia and obesity. The authors also describe that this increased body weight is due to MANF-mediated insulin resistance in the hypothalamus. They suggest that increased MANF enhances PIP4k2b activity in the ER. Based on these findings, the authors conclude that hypothalamic MANF plays a critical role in the regulation of overall energy balance. Although this is a nice study, my enthusiasm for this study is dampened by some major issues described below. In addition, the cellular mechanisms underlying the orexigenic

effect of MANF described in the manuscript are not consistent with the prior studies. In fact, MANF is upregulated by ER stress and protects cells against ER stress-induced cell death as described in the introduction.

Moreover, hypothalamic ER stress appears to be a causative factor for the development of leptin resistance and consequently obesity (Schneeberger et al. 2013, Cell). In other words, increased MANF may counteract ER stress-induced obesity.

Major comments:

1. Figure 1 shows expression of MANF under physiological conditions. However, results are not consistent with what they describe in the text. In fact, it looks like that expression of MANF is much higher in the hypothalamus than in the cortex (1a). However, western blot analysis shows the opposite (1b). In both cases (fed and fasted), the expression of MANF protein appears to be higher in the cortex than in the hypothalamus (1b). In addition, levels of MANF expression in the ARC and LH after fasting (1d) are very similar to those shown in Fig. 1a. Hence, it is unclear whether the hypothalamus is a major area that expresses MANF and whether the expression of MANF is actually regulated by the availability of nutrients.

In **Fig. 1b**, the blots for the cortex and hypothalamus were separated, and developed for different time durations. The same holds true to **Fig. 1a** and **1d**, which were separate experiments developed in 3,3'-Diaminobenzidine (DAB) for different time durations. Therefore, it would be difficult to directly compare the relative levels of proteins between those results. We now provide new western blotting result which includes MANF expression in the cortex, hypothalamus and hippocampus on the same blot. This result clearly indicates that MANF expression is highest in the hypothalamus of wild type mice (**new Supplementary Fig. 1b**), which is in agreement with **Fig. 1a**.

2. Figure 2 and 3 show effects of transgenic MANF expression in the brain. Transgenic mice show hyperphagia and obesity. Importantly, transgenic animals that are pair-fed to WT littermates show no increase in body weight, suggesting that altered food intake results in obesity in transgenic mice. Although data are nice, one needs to know if injection of MANF into the hypothalamus can also promote feeding.

In **Fig. 4**, we injected AAV-MANF into the hypothalamus of wild type mice, and found robust increase in food intake. Since we have generated recombinant mouse MANF

protein, we performed additional experiment to inject MANF protein into the third ventricle. The injected recombinant MANF is likely to act on cells as extracellular proteins. Compared with control mice injected with PBS, mice injected with MANF protein showed similar levels of food intake (**new Supplementary Fig. 4c**), suggesting that extracellular MANF does not regulate feeding. This result is consistent with our mechanistic study that MANF functions intracellularly (in the ER) to modulate hypothalamic insulin signaling and food intake activities. We added the following text to the Discussion:

“Brain injection of recombinant MANF protein has been previously shown neuroprotective in a Parkinson disease rat model. We injected similarly purified recombinant MANF protein into the third ventricle of WT mice, but did not observe any significant changes in food intake. This result suggests that MANF only functions intracellularly to modulate food intake. However, we cannot rule out the possibility that the recombinant MANF protein purified from bacteria lacks certain post-translational modifications that might be essential for extracellular MANF to regulate food intake.”

3. Figure 4 describes effects of MANF overexpression in the hypothalamus. According to their description, AAV vectors were injected into the ventromedial hypothalamus (AP = - 1.5 mm, ML = ± 0.35 mm, DV = - 5.5 mm). If this is the case, MANF expression should be restricted in the area where viruses are injected such as the VMH but not throughout the hypothalamus. The authors do not provide detailed information as to their AAV vectors (titer, serotype, volume, etc). This issue appears to be important to understand how MANF regulates food intake and body weight. In fact, the hypothalamus contains diverse types of neurons such as anorexigenic and orexigenic cells. Since there are NPY, POMC, TH, orexin, and MCH neurons in the ARC and LH, one needs to know what neurons in the ARC, LH and DMH actually express MANF. Based on those results, they can inject viruses specifically into the ARC or the LH.

MANF is expressed in various regions of the hypothalamus (**Fig. 1a**) and may work on several types of neurons in the hypothalamus to regulate feeding behavior. Thus, we injected viral MANF (AAV9, 1.5×10^{13} vg/ml, bilateral injection, 1 μ l in each side) to allow ubiquitous expression of viral MANF in several hypothalamic regions (**Fig. 4c**). We added the information regarding the viruses to the Supplemental Methods. As for which type of neuron in the hypothalamus is more critical for the function of MANF and how different types of hypothalamic neurons participate in MANF-mediated feeding activity, it would require an independent and in-depth study. For example, transgenic expression of MANF

under specific promoters in the selective neuronal types will rigorously address the above issue, which is planned in our future experiments.

4. In page 10 (line 217-220), the authors describe a reduction of POMC expression in mice injected with AAV-MANF due to insulin resistance. Interestingly, the study of Konner and colleagues (2007) shows that insulin signaling in POMC neurons is not involved in feeding behavior. Although all the data described in the manuscript suggest the orexigenic effect of MANF, the cellular mechanism, by which MANF promotes feeding, are not discussed in the text.

It is possible that MANF works on several types of neurons in the hypothalamus to regulate feeding behavior. We included the POMC western blotting result to demonstrate that MANF overexpression indeed caused insulin resistance, given the fact that insulin increases the expression of POMC^{8,9}. We added the following text to the Discussion:

“The CNS contains diverse types of anorexigenic and orexigenic neurons, which respond differently to insulin signaling¹⁰⁻¹². It remains to be investigated which types of neurons express MANF, and how MANF functions in these neurons to mediate their activities.”

5. The authors conclude that increased food intake and body weight in mice overexpressing MANF in the hypothalamus are mediated by insulin resistance. It is true that AKT plays an important role in the insulin-signaling pathways. However, it is not clear if direct injection of insulin into the hypothalamus of mice overexpressing MANF in the hypothalamus has no effect on food intake due to insulin resistance.

We injected insulin into the third ventricle of wild type and MANF TG mice, and measured their daily food intake before and after insulin injection. WT mice had a 60% reduction of daily food intake after insulin injection, whereas MANF transgenic mice had a 30% reduction, which was statistically significant (**new Supplementary Fig. 6a**). Insulin may act on multiple signaling pathways to reduce food intake, but the reduced effect of insulin in MANF transgenic mice also supports the idea that MANF TG mice are insulin resistant.

6. Finally, the cellular mechanisms underlying the orexigenic effects of MANF described in the manuscript are not consistent with the prior studies. In fact, MANF is upregulated by ER stress and protects cells against ER stress-induced cell death. Moreover, hypothalamic ER

stress appears to be a causative factor for the development of leptin resistance and consequently obesity (Schneeberger et al. 2013, Cell). In other words, increased MANF may rather inhibit leptin and insulin resistance.

This is an interesting issue. ER stress can be induced by a variety of intracellular and extracellular factors, and acute and chronic ER stress may differentially impact the function of hypothalamic neurons. The up-regulation of MANF and its protective function during ER stress has been extensively studied by acute treatment, such as ER stress inducing chemicals, or ischemic and epileptic insults^{13,14}. It remains unknown whether MANF is up-regulated under chronic ER stress conditions (such as the development of obesity), and what is the function of MANF during chronic ER stress. The work by Schneeberger et al. (Cell 2013) clearly indicates that Mitofusin 2 in POMC neurons is a molecular link between ER stress and leptin resistance. It is also possible that other molecules in different types of hypothalamic neurons are associated with leptin or insulin resistance in the hypothalamus.

Our original data indicates that MANF overexpression led to insulin resistance in 2-month old MANF TG mice, when the mice were not obese and did not have ER stress in the hypothalamus (**Fig. 6a, Supplementary Fig. 7a**). Here we followed the reviewer's suggestion to test leptin signaling in MANF TG mice. Our new data showed that MANF TG mice before the onset of obesity had comparable levels of plasma leptin as wild type mice, and intraperitoneal injection of leptin led to similar levels of phospho-STAT3 in the hypothalamus of wild type and MANF TG mice (**new Supplementary Fig. 6e, f**). Therefore MANF overexpression is unlikely to affect leptin signaling.

Minor comments:

None

Reviewer #3

Yang et al reported interesting findings that hypothalamic MANF is previously unknown regulator of energy balance. Using various mouse models, the authors convincingly demonstrated that manipulating hypothalamic MANF profoundly affects food intake and body weight. Mechanistically, the authors proposed that MANF suppresses hypothalamic

insulin action, which explains the observed phenotypes. However, brain specific knockout of insulin receptors has modest effects on food intake and body weight. Moreover, there is no data to exclude the possibility that secreted MANF may also promote weight gain independently of insulin. The study is interesting and potentially important, and could be improved by additional experiments to clarify confusions with regard to the underlying mechanisms.

1. It is important to determine the contribution of secreted MANF to energy balance and weight control. Does icv injection of recombinant MANF (the secreted forms) affect food intake and body weight? Does overexpression of the secreted portion of MANF in the hypothalamus affect energy balance and body weight?

The reviewer raised an interesting question. Here we performed additional experiments, in which we injected recombinant mouse MANF protein into the third ventricle, and monitored the feeding activity of the injected mice. Compared with control mice injected with PBS, mice injected with MANF protein showed similar levels of food intake and body weight (**new Supplementary Fig. 4c, d**), indicating that extracellular MANF does not regulate feeding. This result is consistent with our mechanistic study that MANF functions intracellularly (in the ER) to modulate hypothalamic insulin signaling and food intake activities. We added the following text to the Discussion:

“Brain injection of recombinant MANF protein has been previously shown neuroprotective in a Parkinson disease rat model. We injected similarly purified recombinant MANF protein into the third ventricle of WT mice, but did not observe any significant changes in food intake. This result suggests that MANF only functions intracellularly to modulate food intake. However, we cannot rule out the possibility that the recombinant MANF protein purified from bacteria lacks certain post-translational modifications that might be essential for extracellular MANF to regulate food intake.”

2. It is premature to conclude that MANF regulates food intake via suppressing hypothalamic insulin signaling. Does MANF affect leptin signaling? Plasma leptin levels and hypothalamic leptin signaling should be examined in MANF-expressing and knockdown mice before the onset of obesity.

We checked plasma leptin levels in 6-week old wild type and MANF TG mice, when their body weights are still not divergent. These mice had similar levels of plasma leptin (**new Supplementary Fig. 6e**). We also did intraperitoneal injection of leptin to 6-week old

wild type and MANF TG mice, and found comparable levels of phospho-STAT3 in the hypothalamus of injected mice (**new Supplementary Fig. 6f**). Therefore MANF overexpression does not appear to affect leptin signaling.

3. Do hypothalamic expression and ER localization of PIP4k2b and MANF correlate with obesity development?

We used 2-month old MANF TG mice to check the ER localization of PIP4k2b and transgenic MANF (**Fig. 7g, h**). Both proteins are co-increased in the ER in the 2-month old MANF TG mice before these mice develop significant obesity (**Fig. 2e**). We added the age of mice to the revised text to clarify this issue.

4. Hypothalamic insulin is known to regulate systemic glucose metabolism and liver lipid metabolism. Plasma glucose levels, GTT, ITT, and liver steatosis should be assessed in MANF overexpressing or silencing mice.

We performed these experiments suggested by the reviewer using 6-week old MANF TG mice, before their onset of obesity. The results show elevated blood glucose levels, impairment of GTT and ITT, and age related accumulation of fat in the liver of MANF TG mice, compared with their wild type littermates (**new Fig. 6e-g, Supplementary Fig. 6b**). Thus, hypothalamic insulin dysfunction affected systemic glucose metabolism and liver lipid metabolism in MANF TG mice.

References

1. Bradford, J., *et al.* Expression of mutant huntingtin in mouse brain astrocytes causes age-dependent neurological symptoms. *Proceedings of the National Academy of Sciences of the United States of America* **106**, 22480-22485 (2009).
2. Schilling, G., *et al.* Intranuclear inclusions and neuritic aggregates in transgenic mice expressing a mutant N-terminal fragment of huntingtin. *Human molecular genetics* **8**, 397-407 (1999).
3. Emerling, B.M., *et al.* Depletion of a putatively druggable class of phosphatidylinositol kinases inhibits growth of p53-null tumors. *Cell* **155**, 844-857 (2013).
4. Itoh, T., Ijuin, T. & Takenawa, T. A novel phosphatidylinositol-5-phosphate 4-kinase (phosphatidylinositol-phosphate kinase IIgamma) is phosphorylated in the

- endoplasmic reticulum in response to mitogenic signals. *The Journal of biological chemistry* **273**, 20292-20299 (1998).
5. Sarkes, D. & Rameh, L.E. A novel HPLC-based approach makes possible the spatial characterization of cellular PtdIns5P and other phosphoinositides. *The Biochemical journal* **428**, 375-384 (2010).
 6. Carricaburu, V., *et al.* The phosphatidylinositol (PI)-5-phosphate 4-kinase type II enzyme controls insulin signaling by regulating PI-3,4,5-trisphosphate degradation. *Proceedings of the National Academy of Sciences of the United States of America* **100**, 9867-9872 (2003).
 7. Lamia, K.A., *et al.* Increased insulin sensitivity and reduced adiposity in phosphatidylinositol 5-phosphate 4-kinase beta^{-/-} mice. *Molecular and cellular biology* **24**, 5080-5087 (2004).
 8. Benoit, S.C., *et al.* The catabolic action of insulin in the brain is mediated by melanocortins. *The Journal of neuroscience : the official journal of the Society for Neuroscience* **22**, 9048-9052 (2002).
 9. Plum, L., Belgardt, B.F. & Bruning, J.C. Central insulin action in energy and glucose homeostasis. *The Journal of clinical investigation* **116**, 1761-1766 (2006).
 10. Konner, A.C., *et al.* Insulin action in AgRP-expressing neurons is required for suppression of hepatic glucose production. *Cell metabolism* **5**, 438-449 (2007).
 11. Konner, A.C., *et al.* Role for insulin signaling in catecholaminergic neurons in control of energy homeostasis. *Cell metabolism* **13**, 720-728 (2011).
 12. Figlewicz, D.P., Bennett, J.L., Aliakbari, S., Zavosh, A. & Sipols, A.J. Insulin acts at different CNS sites to decrease acute sucrose intake and sucrose self-administration in rats. *American journal of physiology. Regulatory, integrative and comparative physiology* **295**, R388-394 (2008).
 13. Apostolou, A., Shen, Y., Liang, Y., Luo, J. & Fang, S. Armet, a UPR-upregulated protein, inhibits cell proliferation and ER stress-induced cell death. *Experimental cell research* **314**, 2454-2467 (2008).
 14. Lindholm, P., *et al.* MANF is widely expressed in mammalian tissues and differently regulated after ischemic and epileptic insults in rodent brain. *Molecular and cellular neurosciences* **39**, 356-371 (2008).

Reviewers' comments:

Reviewer #1 (Remarks to the Author):

The authors have almost fully addressed my concerns. However, there are two issues that need to be clarified.

1. In Figure 7g, the level of PIP4k2b in the ER fraction was greatly increased in MANF transgenic mice, while the levels of PIP4k2b in other organelles, such as mitochondria, were not changed. I want to know where the increased MANF (in ER) comes from if the authors claimed that MANF expression did not affect the overall level of PIP4k2b in the cells.
2. Also in Figure 7g, the cytoplasm should be changed to cytosol.

Reviewer #2 (Remarks to the Author):

Although the authors addressed some issues, there are still major flaws in the manuscript.

1. The authors claim that MANF is expressed in "various" regions of the hypothalamus. However, this doesn't appear to be the case. Fig. 1 and suppl. Fig. 1 show its expression only in the ARC and the LH. It looks like that those are the areas that express MANF. The authors must show exactly where MANF is expressed within the hypothalamus. This is very important. In fact, the authors suggest that hypothalamic neurons play a major role in MANF-induced increase in food intake and body weight. Yes, it's true that the hypothalamus contains multiple types of neurons. For this reason, they can't inject a very large volume (1 μ l per each side) of virus into the parenchyma of the hypothalamus at a speed of 200 nl per minute. In fact, not only the hypothalamus but also other brain areas would be infected by this treatment (large volume, high speed, and high titer). In other words, increased food intake and body weight following viral injection may not be due to overexpression of MANF in the hypothalamus. To provide evidence of how MANF regulates food intake, they need to inject AAV-MANF into the ARC or the LH. In fact, MANF appears to be expressed mainly in those nuclei. Moreover, the authors describe a reduction of POMC expression in mice injected with AAV-MANF.

They also claim that injection of MANF has no effect on food intake. They need to examine whether injection of MANF (at least two different doses) into the ARC or the LH has no effect. How do they know the concentrations of MANF are not sufficient to reduce food intake? How do they know MANF activates membrane receptors? There is no evidence in the manuscript.

2. They still suggest that increased food intake and body weight in mice overexpressing MANF in the hypothalamus are mediated by insulin resistance, although injection of insulin significantly reduces food intake in MANF transgenic animals (\sim 30 % reduction; Suppl. Fig. 6). How do they explain these findings? These findings do not support their conclusion at all.

Reviewer #3 (Remarks to the Author):

The authors have performed additional experiments and fully addressed my comments. The paper has been strengthened by the new data. I do not have additional questions.

Reviewer #1 (Remarks to the Author):

The authors have almost fully addressed my concerns. However, there are two issues that need to be clarified.

1. In Figure 7g, the level of PIP4k2b in the ER fraction was greatly increased in MANF transgenic mice, while the levels of PIP4k2b in other organelles, such as mitochondria, were not changed. I want to know where the increased MANF (in ER) comes from if the authors claimed that MANF expression did not affect the overall level of PIP4k2b in the cells.

We thank the reviewer for the encouraging comments. In fact, PIP4K2b is decreased in the mitochondria fraction of MANF transgenic mice, especially considering the decreased ratio of PIP4K2b to the mitochondria marker protein COX IV (Fig. 7g), which supports our conclusion that MANF can promote the redistribution of PIP4K2b to the ER. In addition, the sample from the cytosol fraction has a large volume, whereas the ER fraction is highly concentrated. Thus, the redistribution of a fraction of PIP4k2b to the ER may not be readily reflected by its cytosolic level change, but can alter its distribution in other organelles such as mitochondria.

2. Also in Figure 7g, the cytoplasm should be changed to cytosol.

We changed cytoplasm to cytosol.

Reviewer #2 (Remarks to the Author):

Although the authors addressed some issues, there are still major flaws in the manuscript.

1. The authors claim that MANF is expressed in “various” regions of the hypothalamus.

However, this doesn't appear to be the case. Fig. 1 and suppl. Fig. 1 show its expression only in

the ARC and the LH. It looks like that those are the areas that express MANF. The authors must show exactly where MANF is expressed within the hypothalamus. This is very important. In fact, the authors suggest that hypothalamic neurons play a major role in MANF-induced increase in food intake and body weight. Yes, it's true that the hypothalamus contains multiple types of neurons. For this reason, they can't inject a very large volume (1 μ l per each side) of virus into the parenchyma of the hypothalamus at a speed of 200 nl per minute. In fact, not only the hypothalamus but also other brain areas would be infected by this treatment (large volume, high speed, and high titer). In other words, increased food intake and body weight following viral injection may not be due to overexpression of MANF in the hypothalamus. To provide evidence of how MANF regulates food intake, they need to inject AAV-MANF into the ARC or the LH. In fact, MANF appears to be expressed mainly in those nuclei. Moreover, the authors describe a reduction of POMC expression in mice injected with AAV-MANF.

They also claim that injection of MANF has no effect on food intake. They need to examine whether injection of MANF (at least two different doses) into the ARC or the LH has no effect. How do they know the concentrations of MANF are not sufficient to reduce food intake? How do they know MANF activates membrane receptors? There is no evidence in the manuscript.

We appreciate the reviewer's comments on this important issue. Although MANF is highly expressed in the ARC and LH, it is abundantly expressed in other hypothalamic regions as well, including VMH, DMH and PVN. A previous publication also reported high levels of *Manf* mRNA throughout the hypothalamus (Mol Cell Neurosci. 2008 Nov;39(3):356-71). In this revision, we provided new images with low magnification to clearly show that MANF is expressed in different hypothalamic regions such as PVN, LH and ARC (new Supplementary Figure 1a). In fact, due to the broad expression of MANF in the hypothalamus, it appears most appropriate for us to use AAV-UBC-MANF to ubiquitously over-express MANF in the hypothalamus. The high genomic titer does not always translate to high infectious titer. Before conducting the formal experiment,

we tried different injection parameters, including various injection sites and different AAV vector concentrations and volumes, to optimize the condition for injection that allowed for viral transduction in most parts of the hypothalamus without infecting other brain regions. Here we performed western blotting analysis using dissected brain tissues to show that the expression of viral MANF was restricted to the hypothalamus, but not other neighboring brain regions (new Figure 4b). We agree with the reviewer that it is important to understand the function of MANF in different hypothalamic regions, such as the ARC and LH. We were unable to identify literature that could restrict the viral expression to the ARC or LH without using more specific promoters. Nonetheless, we followed the reviewer's suggestion and injected a lower dose (0.2 μ l) of AAV-UBC-MANF into the ARC or LH (new Supplementary Figure 4c). We found that Injection into the ARC, but not the LH, led to a modest increase in food intake and body weight, but this increase was significantly less than that with whole hypothalamus infection (new Supplementary Figure 4d, e). This result suggests that the ARC and other hypothalamic regions are required for the robust effect of MANF on food intake and body weight. We have included this explanation in the revised text.

Regarding recombinant MANF protein injection, we chose the dose of 10 μ g based on a previous publication (J Neurosci. 2009 Jul 29;29(30):9651-9). The membrane receptor for MANF remains unknown, making it difficult to determine the optimal dose. Nonetheless, the previous publication reported that injection of 10 μ g MANF protein yielded maximum protection against neuronal toxicity induced by 6-OHDA. We chose icv injection as requested by Reviewer 3. In this revision, we directly injected a higher dose of MANF protein (8 μ g in each side, 16 μ g in total) into the hypothalamus. Because of the high dose injected, it is impossible to precisely restrict the distribution of MANF protein to certain hypothalamic regions. Therefore we used the same stereotaxic coordinates and performed the direct injection of a higher dose of MANF protein into the hypothalamus, as requested by the reviewer. The new result still showed no

effect of extracellular MANF on food intake (new Supplementary Figure 4g), which is consistent with the conclusion of our study.

2. They still suggest that increased food intake and body weight in mice overexpressing MANF in the hypothalamus are mediated by insulin resistance, although injection of insulin significantly reduces food intake in MANF transgenic animals (~ 30 % reduction; Suppl. Fig. 6). How do they explain these findings? These findings do not support their conclusion at all.

The extent for insulin resistance is dependent on the dose of administered insulin. For supplemental Figure 6a, we performed acute icv injection of a high dose of insulin (1 μ U), which does not occur under physiological conditions. Under this experimental condition, WT mice showed ~ 60% food intake reduction, whereas MANF transgenic mice showed only ~30% reduction. The attenuated reduction strongly indicates that the hypothalamus of MANF transgenic mice is less sensitive to insulin. Under physiological conditions, the hypothalamic insulin resistance in MANF transgenic mice could be more significant and lead to hyperphagia and obesity, which has been supported by other data in our study.

Reviewer #3 (Remarks to the Author):

The authors have performed additional experiments and fully addressed my comments. The paper has been strengthened by the new data. I do not have additional questions.

We thank the reviewer for the encouraging comments.

REVIEWERS' COMMENTS:

Reviewer #2 (Remarks to the Author):

Although the authors addressed some issues, there are still major flaws in the manuscript.

Re: 1. Although MANF is highly expressed in the ARC and LH, it is abundantly expressed in other hypothalamic regions as well, including VMH, DMH and PVN. In this revision, we provided new images with low magnification to clearly show that MANF is expressed in different hypothalamic regions such as PVN, LH and ARC (new Supplementary Figure 1a). Due to the broad expression of MANF in the hypothalamus, it appears most appropriate for us to use AAV-UBC-MANF to ubiquitously over-express MANF in the hypothalamus.

- . New Suppl. Fig. 1a shows expression of MANF in the PVN, LH, and ARC, not other hypothalamic areas. The authors claimed that they used the UBC promoter to "ubiquitously" express MANF in the hypothalamus. However, there are still no images that actually show the "ubiquitous" expression of MANF in the hypothalamus. Does the injection of this virus actually drive the expression in the PVN, LH, and ARC at least?

In addition, it is not clear at all why mice injected with this AAV into the hypothalamus gain more body weight than MANF transgenic mice do. This would be due to VMH lesions produced by a large-volume high speed injection of AAV into the VMH. Male mice gain ~10g only 2 weeks post viral injections (Fig. 4), which is very similar to that observed in mice following VMH lesions. The authors should do immunostaining with an anti-HA antibody to show the expression of MANF in the hypothalamus and brain. Why did they do only western blotting?

Although mice injected with AAV and MANF transgenic mice are obese, there is no increase in plasma leptin levels (Suppl. Fig. 6). Why? Is there an increase in plasma insulin levels in MANF transgenic mice and mice injected with AAV? The authors claimed that there was impaired GTT in TG mice, but it seems that there was only a minor difference between TG and control mice (Fig. 6).

Re. 2. The extent for insulin resistance is dependent on the dose of administered insulin. For supplemental Figure 6a, we performed acute icv injection of a high dose of insulin (1 μ U), which does not occur under physiological conditions. Under this experimental condition, WT mice showed ~ 60% food intake reduction, whereas MANF transgenic mice showed only ~30% reduction. The attenuated reduction strongly indicates that the hypothalamus of MANF transgenic mice is less sensitive to insulin. Under physiological conditions, the hypothalamic insulin resistance in MANF transgenic mice could be more significant and lead to hyperphagia and obesity, which has been supported by other data in our study.

- . If so, the authors should use physiological concentrations of insulin and provide results. As described above, the authors did not even show plasma insulin levels in TG and mice injected with AAV.

Response to referees

Although the authors addressed some issues, there are still major flaws in the manuscript.

Re: 1. Although MANF is highly expressed in the ARC and LH, it is abundantly expressed in other hypothalamic regions as well, including VMH, DMH and PVN. In this revision, we provided new images with low magnification to clearly show that MANF is expressed in different hypothalamic regions such as PVN, LH and ARC (new Supplementary Figure 1a). Due to the broad expression of MANF in the hypothalamus, it appears most appropriate for us to use AAV-UBC-MANF to ubiquitously overexpress MANF in the hypothalamus.

- New Suppl. Fig. 1a shows expression of MANF in the PVN, LH, and ARC, not other hypothalamic areas. The authors claimed that they used the UBC promoter to “ubiquitously” express MANF in the hypothalamus. However, there are still no images that actually show the “ubiquitous” expression of MANF in the hypothalamus. Does the injection of this virus actually drive the expression in the PVN, LH, and ARC at least?

In Supplemental Fig 1a, we provided a clear image showing that MANF is expressed in different nuclei of the hypothalamus, with the highest expression in the PVN, LH and ARC. Previous published papers also indicate MANF is ubiquitously expressed in the hypothalamus (Mol Cell Neurosci. 2008 Nov;39(3):356-71), and we have indicated this in the text by citing the published paper. In order to address the previous issues regarding the distribution of AAV-MANF in different nuclei in the hypothalamus, we performed immunostaining that is able to clearly show the distribution of AAV-MANF in distinct areas in the hypothalamus. As shown in Fig. 4c, AAV-MANF is indeed expressed in ARC, DMH and LH at least. This figure had been presented in the previous versions multiple times.

In addition, it is not clear at all why mice injected with this AAV into the hypothalamus gain more body weight than MANF transgenic mice do. This would be due to VMH lesions produced by a large-volume high speed injection of AAV into the VMH. Male mice gain ~10g only 2 weeks post viral injections (Fig. 4), which is very similar to that observed in mice following VMH lesions. The authors should do immunostaining with an anti-HA antibody to show the expression of MANF in the hypothalamus and brain. Why did they do only western blotting?

The difference between the AAV-MANF injected mice and the MANF transgenic suggests a dose-dependent effect. AAV injection leads to a much higher expression level of MANF than transgene expression in MANF transgenic mice. The difference also underscores the importance of hypothalamic specific expression of MANF in regulating body weight, rather than MANF expression in other tissues due to germline expression of the transgenic MANF.

If VMH lesions caused the body weight change, we should see significant increases in body weight of AAV-GFP injected control mice, which were injected under the same experimental conditions as AAV-MANF injection. However, this is not the case when we compared AAV-GFP injection and non-injected mice (data not shown). Instead, AAV-MANF

injection dramatically increased body weight compared with AAV-GFP injection (Fig. 4e). In addition, we also knocked down MANF in the hypothalamus using AAV viruses with the same injection parameters, and found reduced food intake and body weight (Fig. 5). This is opposite to the increased food intake and body weight caused by AAV-MANF injection. All the results clearly indicate that body weight changes are not caused by VMH lesions.

As for the examination of AAV-MANF expression in the hypothalamus, we did use anti-HA immunohistochemistry to show AAV-MANF expression in the hypothalamus in Fig. 4c. In this Figure, high magnification micrographs would allow us to provide clear evidence for the expression of AAV-MANF in different nuclei in the hypothalamus. The reviewer previously had concerns about the virus leaking to other brain regions, which is why we added another solid evidence by including western blotting results showing that AAV-MANF is only expressed in the hypothalamus (Fig. 4b), as western blotting can clearly reveal the right size of the injected AAV-MANF that is only presented in the injected hypothalamic area compared with other brain regions.

Although mice injected with AAV and MANF transgenic mice are obese, there is no increase in plasma leptin levels (Suppl. Fig. 6). Why? Is there an increase in plasma insulin levels in MANF transgenic mice and mice injected with AAV? The authors claimed that there was impaired GTT in TG mice, but it seems that there was only a minor difference between TG and control mice (Fig. 6).

The main point of our paper is that MANF overexpression causes insulin resistance, which leads to hyperphagia and obesity. It is well established that obesity could lead to leptin and insulin resistance (J Clin Invest. 2000 Aug 15; 106(4): 473–481; Trends Endocrinol Metab. 2010 Nov; 21(11): 643–651.). Therefore, to convincingly address the role of MANF overexpression in insulin and leptin signaling, we have to perform all these metabolic tests before the onset of obesity in MANF transgenic mice. Supplemental Fig. 6 shows plasma leptin levels in 6-week old wild type and MANF TG mice, when their body weights were still not divergent, and these mice had similar levels of plasma leptin. Indeed, this question has been specifically requested by Reviewer 3 in the first round of revision: “Plasma leptin levels and hypothalamic leptin signaling should be examined in MANF-expressing and knockdown mice before the onset of obesity.”

Our current data indicate that before the onset of obesity, MANF transgenic mice had impaired insulin response (Fig. 6), but normal leptin response (Supplemental Fig. 6e, f), which supports our conclusion that MANF overexpression leads to insulin resistance, but not leptin resistance. Reviewer 3 has agreed that we successfully addressed this issue. We also have data showing significantly elevated plasma insulin and leptin levels in MANF transgenic mice after the onset of obesity (see below), but this result is expected in obese mice and is thus not included in the paper, as it is not the key issue in our paper.

Figure Legend: Plasma insulin (left) and leptin (right) levels in 6-month old wild type (WT) and MANF transgenic (TG) mice were determined by ELISA (* $P < 0.05$, ** $P < 0.01$, *** $P < 0.001$, $n = 3$; student t test; for insulin, Male, $t = 3.12$, $P = 0.0355$, Female, $t = 3.948$, $P = 0.0168$; for leptin, Male, $t = 5.495$, $P = 0.0053$, Female, $t = 11.83$, $P = 0.0003$).

The impairment of glucose tolerance test (GTT) in pre-obese MANF transgenic mice reached the significant level (Fig. 6f). And a similar extent of impaired GTT has been reported in many other mouse models (Nature. 2001 Mar 8;410(6825):207-12; Proc Natl Acad Sci U S A. 2006 Jan 24;103(4):1006-11). Again, the impairment is much more dramatic in obese MANF transgenic mice, but that result could be confounded by the body composition of the mice and is not directly relevant to the conclusion of our study. Thus, we only presented GTT in pre-obese MANF transgenic mice.

Re. 2. The extent for insulin resistance is dependent on the dose of administered insulin. For supplemental Figure 6a, we performed acute icv injection of a high dose of insulin (1 μ U), which does not occur under physiological conditions. Under this experimental condition, WT mice showed ~ 60% food intake reduction, whereas MANF transgenic mice showed only ~30% reduction. The attenuated reduction strongly indicates that the hypothalamus of MANF transgenic mice is less sensitive to insulin. Under physiological conditions, the hypothalamic insulin resistance in MANF transgenic mice could be more significant and lead to hyperphagia and obesity, which has been supported by other data in our study.

- If so, the authors should use physiological concentrations of insulin and provide results. As described above, the authors did not even show plasma insulin levels in TG and mice injected with AAV.

The physiological concentrations of insulin are present in animals that were examined without any insulin administration, which has already been reflected by the increased food intake in MANF transgenic mice and AAV-MANF injected mice (Fig. 3a, 4f). In addition, previous studies routinely perform icv injection with 0.4 – 4 μ U insulin and examine food intake in the mice after insulin injection (Physiol Behav. 2006 Dec 30;89(5):687-91; Physiol Behav. 2015 Nov 1;151:623-8). This is the reason why we used 1 μ U insulin in our study.

We have data showing significantly increased plasma insulin levels in pre-obese MANF transgenic mice (see below). However, in our previous revision Reviewer 3 specifically requested: “Plasma glucose levels, GTT, ITT, and liver steatosis should be assessed in MANF overexpressing or silencing mice”. We have already added substantial new results to address Reviewer 3 and other reviewers’ issues. Since our current data already strongly support the conclusion that MANF overexpression leads to insulin resistance (Fig. 6e-g), we did not include this figure in the revised paper.

Figure Legend: Plasma insulin levels of 6-week old wild type (WT) and MANF transgenic (TG) mice were measured (* $P < 0.05$, ** $P < 0.01$, $n = 5$; student t test, Male, $t = 2.94$, $P = 0.0187$; Female, $t = 4.307$, $P = 0.0026$).